# Accelerating Hamiltonian Monte Carlo via Chebyshev Integration Time

**Jun-Kun Wang and Andre Wibisono**
Department of Computer Science, Yale University
{jun-kun.wang,andre.wibisono}@yale.edu

## Abstract

Hamiltonian Monte Carlo (HMC) is a popular method in sampling. While there are quite a few works of studying this method on various aspects, an interesting question is how to choose its integration time to achieve acceleration. In this work, we consider accelerating the process of sampling from a distribution $\pi(x) \propto \exp(-f(x))$ via HMC via time-varying integration time. When the potential $f$ is $L$-smooth and $m$-strongly convex, i.e. for sampling from a log-smooth and strongly log-concave target distribution $\pi$, it is known that under a constant integration time, the number of iterations that ideal HMC takes to get an $\epsilon$ Wasserstein-2 distance to the target $\pi$ is $O(\kappa \log \frac{1}{\epsilon})$, where $\kappa := \frac{L}{m}$ is the condition number. We propose a scheme of time-varying integration time based on the roots of Chebyshev polynomials. We show that in the case of quadratic potential $f$, i.e. when the target $\pi$ is a Gaussian distribution, ideal HMC with this choice of integration time only takes $O(\sqrt{\kappa} \log \frac{1}{\epsilon})$ number of iterations to reach Wasserstein-2 distance less than $\epsilon$; this improvement on the dependence on condition number is akin to acceleration in optimization. The design and analysis of HMC with the proposed integration time is built on the tools of Chebyshev polynomials. Experiments find the advantage of adopting our scheme of time-varying integration time even for sampling from distributions with smooth strongly convex potentials that are not quadratic.

## 1 Introduction

Markov chain Monte Carlo (MCMC) algorithms are fundamental techniques for sampling from probability distributions, which is a task that naturally arises in statistics (Duane et al., 1987; Girolami & Calderhead, 2011), optimization (Flaxman et al., 2005; Duchi et al., 2012; Jin et al., 2017), machine learning and others (Wenzel et al., 2020; Salakhutdinov & Mnih, 2008; Koller & Friedman, 2009; Welling & Teh, 2011). Among all the MCMC algorithms, the most popular ones perhaps are Langevin methods (Li et al., 2022; Dalalyan, 2017; Durmus et al., 2019; Vempala & Wibisono, 2019; Lee et al., 2021b; Chewi et al., 2020) and Hamiltonian Monte Carlo (HMC) (Neal, 2012; Betancourt, 2017; Hoffman & Gelman, 2014; Levy et al., 2018). For the former, recently there have been a sequence of works leveraging some techniques in optimization to design Langevin methods, which include borrowing the idea of momentum methods like Nesterov acceleration (Nesterov, 2013) to design fast methods, e.g., (Ma et al., 2021; Dalalyan & Riou-Durand, 2020). Specifically, Ma et al. (2021) show that for sampling from distributions satisfying the log-Sobolev inequality, under-damped Langevin improves the iteration complexity of over-damped Langevin from $O(\frac{d}{\epsilon})$ to $O(\sqrt{\frac{d}{\epsilon}})$, where $d$ is the dimension and $\epsilon$ is the error in KL divergence, though whether their result has an optimal dependency on the condition number is not clear. On the other hand, compared to Langevin methods, the connection between HMCs and techniques in optimization seems rather loose. Moreover, to our knowledge, little is known about how to accelerate HMCs with a provable acceleration guarantee for converging to a target distribution. Specifically, Chen & Vempala (2019) show that for sampling from strongly log-concave distributions, the iteration complexity of ideal HMC is $O(\kappa \log \frac{1}{\epsilon})$, and Vishnoi (2021) shows the same rate of ideal HMC when the potential is strongly convex quadratic in a nice tutorial. In contrast, there are a few methods that exhibit acceleration when minimizing strongly convex quadratic functions in optimization. For example, while Heavy Ball (Polyak, 1964) does not have an accelerated linear rate globally for minimizing general smooth strongly convex functions, it does show acceleration when minimizing strongly convex quadratic functions (Wang et al., 2020;

---

**Algorithm 1:** IDEAL HMC

---

1: Require: an initial point $x_0 \in \mathbb{R}^d$, number of iterations $K$, and a scheme of integration time $\{\eta_k^{(K)}\}$.
2: **for** $k = 1$ to $K$ **do**
3:     Sample velocity $\xi \sim N(0, I_d)$.
4:     Set $(x_k, v_k) = \text{HMC}_{\eta_k^{(K)}}(x_{k-1}, \xi)$.
5: **end for**

---

2021; 2022). This observation makes us wonder whether one can get an accelerated linear rate of ideal HMC for sampling, i.e., $O(\sqrt{\kappa} \log \frac{1}{\epsilon})$, akin to acceleration in optimization.

We answer this question affirmatively, at least in the Gaussian case. We propose a time-varying integration time for HMC, and we show that ideal HMC with this time-varying integration time exhibits acceleration when the potential is a strongly convex quadratic (i.e. the target $\pi$ is a Gaussian), compared to what is established in Chen & Vempala (2019) and Vishnoi (2021) for using a constant integration time. Our proposed time-varying integration time at each iteration of HMC depends on the total number of iterations $K$, the current iteration index $k$, the strong convexity constant $m$, and the smoothness constant $L$ of the potential; therefore, the integration time at each iteration is simple to compute and is set before executing HMC. Our proposed integration time is based on the roots of Chebysev polynomials, which we will describe in details in the next section. In optimization, Chebyshev polynomials have been used to help design accelerated algorithms for minimizing strongly convex quadratic functions, i.e., Chebyshev iteration (see e.g., Section 2.3 in d'Aspremont et al. (2021)). Our result of accelerating HMC via using the proposed Chebyshev integration time can be viewed as the sampling counterpart of acceleration from optimization. Interestingly, for minimizing strongly convex quadratic functions, acceleration of vanilla gradient descent can be achieved via a scheme of step sizes that is based on a Chebyshev polynomial, see e.g., Agarwal et al. (2021), and our work is inspired by a nice blog article by Pedregosa (2021). Hence, our acceleration result of HMC can also be viewed as a counterpart in this sense. In addition to our theoretical findings, we conduct experiments of sampling from a Gaussian as well as sampling from distributions whose potentials are not quadratics, which include sampling from a mixture of two Gaussians, Bayesian logistic regression, and sampling from a *hard* distribution that was proposed in Lee et al. (2021a) for establishing some lower-bound results of certain Metropolized sampling methods. Experimental results show that our proposed time-varying integration time also leads to a better performance compared to using the constant integration time of Chen & Vempala (2019) and Vishnoi (2021) for sampling from the distributions whose potential functions are not quadratic. We conjecture that our proposed time-varying integration time also helps accelerate HMC for sampling from log-smooth and strongly log-concave distributions, and we leave the analysis of such cases for future work.

## 2 PRELIMINARIES

### 2.1 HAMILTONIAN MONTE CARLO (HMC)

Suppose we want to sample from a target probability distribution $\nu(x) \propto \exp(-f(x))$ on $\mathbb{R}^d$, where $f \colon \mathbb{R}^d \to \mathbb{R}$ is a continuous function which we refer to as the potential.

Denote $x \in \mathbb{R}^d$ the position and $v \in \mathbb{R}^d$ the velocity of a particle. In this paper, we consider the standard *Hamiltonian* of the particle (Chen & Vempala, 2019; Neal, 2012), which is defined as

$$H(x, v) := f(x) + \tfrac{1}{2}\|v\|^2, \tag{1}$$

while we refer the readers to Girolami & Calderhead (2011); Hirt et al. (2021); Brofos & Lederman (2021) and the references therein for other notions of the Hamiltonian. The *Hamiltonian flow* generated by $H$ is the flow of the particle which evolves according to the following differential equations:

$$\frac{dx}{dt} = \frac{\partial H}{\partial v} \quad \text{and} \quad \frac{dv}{dt} = -\frac{\partial H}{\partial x}.$$

For the standard Hamiltonian defined in (1), the Hamiltonian flow becomes

$$\frac{dx}{dt} = v \quad \text{and} \quad \frac{dv}{dt} = -\nabla f(x). \tag{2}$$

We will write $(x_t, v_t) = \text{HMC}_t(x_0, v_0)$ as the position $x$ and the velocity $v$ of the Hamiltonian flow after integration time $t$ starting from $(x_0, v_0)$. There are many important properties of the Hamiltonian flow including that the Hamiltonian is conserved along the flow, the vector field associated with the flow is divergence free, and the Hamiltonian dynamic is time reversible, see e.g., Section 3 in Vishnoi (2021).

The **Ideal HMC** algorithm (see Algorithm 1) proceeds as follows: in each iteration $k$, sample an initial velocity from the normal distribution, and then flow following the Hamiltonian flow with a pre-specified integration time $\eta_k$. It is well-known that ideal HMC preserves the target density $\pi(x) \propto \exp(-f(x))$; see e.g., Theorem 5.1 in Vishnoi (2021). Furthermore, in each iteration, HMC brings the density of the iterates $x_k \sim \rho_k$ closer to the target $\pi$. However, the Hamiltonian flow $\text{HMC}_t(x_0, v_0)$ is in general difficult to simulate exactly, except for some special potentials. In practice, the Verlet integrator is commonly used to approximate the flow and a Metropolis-Hastings filter is applied to correct the induced bias arises from the use of the integrator (Tripuraneni et al., 2017; Brofos & Lederman, 2021; Hoffman et al., 2021; Lee et al., 2021a; Chen et al., 2020). In recent years, there have been some progress on showing some rigorous theoretical guarantees of HMCs for converging to a target distribution, e.g., Chen et al. (2020); Durmus et al. (2017); Bou-Rabee & Eberle (2021); Mangoubi & Smith (2019; 2021); Mangoubi & Vishnoi (2018). There are also other variants of HMCs proposed in the literature, e.g., Riou-Durand & Vogrinc (2022); Bou-Rabee & Sanz-Serna (2017); Zou & Gu (2021); Steeg & Galstyan (2021); Hoffman & Gelman (2014); Tripuraneni et al. (2017); Chen et al. (2014), to name just a few.

Recall that the 2-Wasserstein distance between probability distributions $\nu_1$ and $\nu_2$ is

$$\text{W}_2(\nu_1, \nu_2) := \inf_{x,y \in \Gamma(\nu_1, \nu_2)} \mathbb{E}\left[\|x - y\|^2\right]^{1/2}$$

where $\Gamma(\nu_1, \nu_2)$ represents the set of all couplings of $\nu_1$ and $\nu_2$.

## 2.2 ANALYSIS OF HMC IN QUADRATIC CASE WITH CONSTANT INTEGRATION TIME

In the following, we replicate the analysis of ideal HMC with a constant integration time for quadratic potentials (Vishnoi, 2021), which provides the necessary ingredients for introducing our method in the next section. Specifically, we consider the following quadratic potential:

$$f(x) := \sum_{j=1}^{d} \lambda_j x_j^2, \text{ where } 0 < m \le \lambda_j \le L, \tag{3}$$

which means the target density is the Gaussian distribution $\pi = \mathcal{N}(0, \Lambda^{-1})$, where $\Lambda$ the diagonal matrix whose $j^{\text{th}}$ diagonal entry is $\lambda_j$. We note for a general Gaussian target $\mathcal{N}(\mu, \Sigma)$ for some $\mu \in \mathbb{R}^d$ and $\Sigma \succ 0$, we can shift and rotate the coordinates to make $\mu = 0$ and $\Sigma$ a diagonal matrix, and our analysis below applies. So without loss of generality, we may assume the quadratic potential is separable, as in (3).

In this quadratic case, the Hamiltonian flow (2) becomes a linear system of differential equations, and we have an exact solution given by sinusoidal functions, which are

$$\begin{aligned} x_t[j] &= \cos\left(\sqrt{2\lambda_j}t\right) x_0[j] + \frac{1}{\sqrt{2\lambda_j}} \sin\left(\sqrt{2\lambda_j}t\right) v_0[j], \\ v_t[j] &= -\sqrt{2\lambda_j} \sin\left(\sqrt{2\lambda_j}t\right) x_0[j] + \cos\left(\sqrt{2\lambda_j}t\right) v_0[j]. \end{aligned} \tag{4}$$

In particular, we recall the following result on the deviation between two co-evolving particles with the same initial velocity.

**Lemma 1.** *(Vishnoi, 2021) Let $x_0, y_0 \in \mathbb{R}^d$. Consider the following coupling: $(x_t, v_t) = \text{HMC}_t(x_0, \xi)$ and $(y_t, u_t) = \text{HMC}_t(y_0, \xi)$ for some $\xi \in \mathbb{R}^d$. Then for all $t \ge 0$ and for all $j \in [d]$, it holds that*

$$x_t[j] - y_t[j] = \cos\left(\sqrt{2\lambda_j}t\right) \times (x_0[j] - y_0[j]).$$

Using Lemma 1, we can derive the convergence rate of ideal HMC for the quadratic potential as follows.

**Lemma 2.** *(Vishnoi, 2021) Let $\pi \propto \exp(-f) = \mathcal{N}(0, \Lambda^{-1})$ be the target distribution, where $f(x)$ is defined on (3). Let $\rho_K$ be the distribution of $x_K$ generated by Algorithm 1 at the final iteration $K$. Then for any $\rho_0$ and any $K \geq 1$, we have*

$$W_2(\rho_K, \pi) \leq \max_{j \in [d]} \left| \Pi_{k=1}^K \cos\left(\sqrt{2\lambda_j} \eta_k^{(K)}\right) \right| W_2(\rho_0, \pi).$$

We replicate the proof of Lemma 1 and Lemma 2 in Appendix B for the reader's convenience.

Vishnoi (2021) shows that by choosing

$$\text{(\textbf{Constant integration time})} \qquad \eta_k^{(K)} = \frac{\pi}{2} \frac{1}{\sqrt{2L}}, \tag{5}$$

one has that $\cos\left(\sqrt{2\lambda_j} \eta_k^{(K)}\right) \leq 1 - \Theta\left(\frac{m}{L}\right)$ for all the iterations $k \in [K]$ and dimensions $j \in [d]$. Hence, by Lemma 2, the distance satisfies

$$W_2(\rho_K, \pi) = O\left(\left(1 - \Theta\left(\frac{m}{L}\right)\right)^K\right) W_2(\rho_0, \pi)$$

after $K$ iterations of ideal HMC with the constant integration time. On the other hand, for general smooth strongly convex potentials $f(\cdot)$, Chen & Vempala (2019) show the same convergence rate $1 - \Theta\left(\frac{m}{L}\right)$ of HMC using a constant integration time $\eta_k^{(K)} = \frac{c}{\sqrt{L}}$, where $c > 0$ is a universal constant. Therefore, under the constant integration time, HMC needs $O(\kappa \log \frac{1}{\epsilon})$ iterations to reach error $W_2(\rho_K, \pi) \leq \epsilon$, where $\kappa = \frac{L}{m}$ is condition number. Furthermore, they also show that the relaxation time of ideal HMC with a constant integration time is $\Omega(\kappa)$ for the Gaussian case.

## 2.3 CHEBYSHEV POLYNOMIALS

We denote $\Phi_K(\cdot)$ the degree-$K$ *Chebyshev polynomial of the first kind*, which is defined by:

$$\Phi_K(x) = \begin{cases} \cos(K \arccos(x)) & \text{if } x \in [-1, 1], \\ \cosh(K \operatorname{arccosh}(x)) & \text{if } x > 1, \\ (-1)^K \cosh(K \operatorname{arccosh}(x)) & \text{if } x < 1. \end{cases} \tag{6}$$

Our proposed integration time is built on a scaled-and-shifted Chebyshev polynomial, defined as:

$$\bar{\Phi}_K(\lambda) := \frac{\Phi_K(h(\lambda))}{\Phi_K(h(0))}, \tag{7}$$

where $h(\cdot)$ is the mapping $h(\lambda) := \frac{L+m-2\lambda}{L-m}$. Observe that the mapping $h(\cdot)$ maps all $\lambda \in [m, L]$ into the interval $[-1, 1]$. The roots of the degree-$K$ scaled-and-shifted Chebyshev polynomial $\bar{\Phi}_K(\lambda)$ are

$$\text{(\textbf{Chebyshev roots})} \qquad r_k^{(K)} := \frac{L+m}{2} - \frac{L-m}{2} \cos\left(\frac{(k-\frac{1}{2})\pi}{K}\right), \tag{8}$$

where $k = 1, 2, \ldots, K$, i.e., $\bar{\Phi}_K(r_k^{(K)}) = 0$. We now recall the following key result regarding the scaled-and-shifted Chebyshev polynomial $\bar{\Phi}_K$.

**Lemma 3.** *(e.g., Section 2.3 in d'Aspremont et al. (2021)) For any positive integer $K$, we have*

$$\max_{\lambda \in [m, L]} \left| \bar{\Phi}_K(\lambda) \right| \leq 2 \left(1 - 2\frac{\sqrt{m}}{\sqrt{L}+\sqrt{m}}\right)^K = O\left(\left(1 - \Theta\left(\sqrt{\frac{m}{L}}\right)\right)^K\right). \tag{9}$$

The proof of Lemma 3 is in Appendix B.

## 3 CHEBYSHEV INTEGRATION TIME

We are now ready to introduce our scheme of time-varying integration time. Let $K$ be the pre-specified total number of iterations of HMC. Our proposed method will first permute the array $[1, 2, \ldots, K]$ before executing HMC for $K$ iterations. Denote $\sigma(k)$ the $k_{\text{th}}$ element of the array $[1, 2, \ldots, K]$ *after an arbitrary permutation $\sigma$*. Then, we propose to set the integration time of HMC at iteration $k$, i.e., set $\eta_k^{(K)}$, as follows:

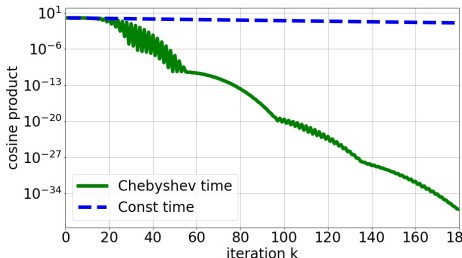 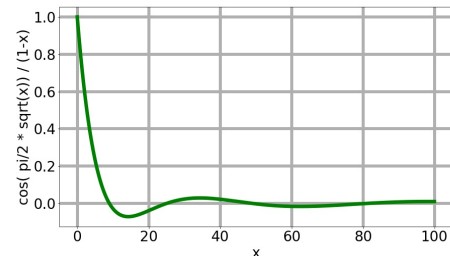

Figure 1: **Left**: Set $K = 400$, $m = 1$ and $L = 100$. The green solid line (Chebyshev integration time (10)) on the subfigure represents $\max_{\lambda \in \{m, m+0.1, \ldots, L\}} \left| \Pi_{s=1}^{k} \cos\left( \sqrt{2\lambda} \eta_s^{(K)} \right) \right| = \left| \Pi_{s=1}^{k} \cos\left( \frac{\pi}{2} \sqrt{\frac{\lambda}{r_{\sigma(s)}^{(K)}}} \right) \right|$ v.s. $k$, while the blue dash line (Constant integration time (5)) represents $\max_{\lambda \in \{m, m+0.1, \ldots, L\}} \left| \Pi_{s=1}^{k} \cos\left( \sqrt{2\lambda} \eta_s^{(K)} \right) \right| = \left| \Pi_{s=1}^{k} \cos\left( \frac{\pi}{2} \sqrt{\frac{\lambda}{L}} \right) \right|$ v.s. $k$. Since the cosine product controls the convergence rate of the $W_2$ distance by Lemma 2, this confirms the acceleration via using the proposed scheme of Chebyshev integration over the constant integration time (Chen & Vempala, 2019; Vishnoi, 2021). **Right:** $\psi(x) = \frac{\cos\left( \frac{\pi}{2} \sqrt{x} \right)}{1-x}$ v.s. $x$.

$$\text{(\textbf{Chebyshev integration time})} \qquad \eta_k^{(K)} = \frac{\pi}{2} \frac{1}{\sqrt{2 r_{\sigma(k)}^{(K)}}}. \tag{10}$$

We note the usage of the permutation $\sigma$ is not needed in our analysis below; however, it seems to help improve performance in practice. Specifically, though the guarantees of HMC at the final iteration $K$ provided in Theorem 1 and Lemma 4 below is the same regardless of the permutation, the progress of HMC varies under different permutations of the integration time, which is why we recommend an arbitrary permutation of the integration time in practice.

Our main result is the following improved convergence rate of HMC under the Chebyshev integration time, for quadratic potentials.

**Theorem 1.** *Denote the target distribution $\pi \propto \exp(-f(x)) = \mathcal{N}(0, \Lambda^{-1})$, where $f(x)$ is defined on (3), and denote the condition number $\kappa := \frac{L}{m}$. Let $\rho_K$ be the distribution of $x_K$ generated by Algorithm 1 at the final iteration $K$. Then, we have*

$$W_2(\rho_K, \pi) \leq 2 \left( 1 - 2 \frac{\sqrt{m}}{\sqrt{L} + \sqrt{m}} \right)^K W_2(\rho_0, \pi) = O\left( \left( 1 - \Theta\left( \frac{1}{\sqrt{\kappa}} \right) \right)^K \right) W_2(\rho_0, \pi).$$

*Consequently, the total number of iterations $K$ such that the Wasserstein-2 distance satisfies $W_2(\rho_K, \pi) \leq \epsilon$ is $O\left( \sqrt{\kappa} \log \frac{1}{\epsilon} \right)$.*

Theorem 1 shows an accelerated linear rate $1 - \Theta\left( \frac{1}{\sqrt{\kappa}} \right)$ using Chebyshev integration time, and hence improves the previous result of $1 - \Theta\left( \frac{1}{\kappa} \right)$ as discussed above. The proof of Theorem 1 relies on the following lemma, which upper-bounds the cosine products that appear in the bound of the $W_2$ distance in Lemma 2 by the scaled-and-shifted Chebyshev polynomial $\bar{\Phi}_K(\lambda)$ on (7).

**Lemma 4.** *Denote $|P_K^{\text{Cos}}(\lambda)| := \left| \Pi_{k=1}^{K} \cos\left( \frac{\pi}{2} \sqrt{\frac{\lambda}{r_{\sigma(k)}^{(K)}}} \right) \right|$. Suppose $\lambda \in [m, L]$. Then, we have for any positive integer $K$,*

$$|P_K^{\text{Cos}}(\lambda)| \leq \left| \bar{\Phi}_K(\lambda) \right|. \tag{11}$$

The proof of Lemma 4 is available in Appendix C. Figure 1 compares the cosine product $\max_{\lambda \in [m, L]} \left| \Pi_{s=1}^{k} \cos\left( \sqrt{2\lambda} \eta_s^{(K)} \right) \right|$ in Lemma 2 of using the proposed integration time and that

---

**Algorithm 2:** HMC WITH CHEBYSHEV INTEGRATION TIME

---

1: Given: a potential $f(\cdot)$, where $\pi(x) \propto \exp(-f(x))$ and $f(\cdot)$ is $L$-smooth and $m$-strongly convex.
2: Require: number of iterations $K$ and the step size of the leapfrog steps $\theta$.
3: Define $r_k^{(K)} := \frac{L+m}{2} - \frac{L-m}{2}\cos\left(\frac{(k-\frac{1}{2})\pi}{K}\right)$, for $k = 1, \ldots, K$.
4: Arbitrarily permute the array $[1, 2, \ldots, K]$. Denote $\sigma(k)$ the $k_{\text{th}}$ element of the array after permutation.
5: **for** $k = 1, 2, \ldots, K$ **do**
6:     Sample velocity $\xi_k \sim N(0, I_d)$.
7:     Set integration time $\eta_k^{(K)} \leftarrow \frac{\pi}{2} \frac{1}{\sqrt{2r_{\sigma(k)}^{(K)}}}$.
8:     Set the number of *leapfrog* steps $S_k \leftarrow \lfloor \frac{\eta_k^{(K)}}{\theta} \rfloor$.
9:     $(\bar{x}_0, \bar{v}_0) \leftarrow (x_{k-1}, \xi_k)$
        % Leapfrog steps
10:    **for** $s = 0, 2, \ldots, S_k - 1$ **do**
11:        $\bar{v}_{s+\frac{1}{2}} = \bar{v}_s - \frac{\theta}{2}\nabla f(\bar{x}_s);$      $\bar{x}_{s+1} = \bar{x}_s + \theta\bar{v}_{s+\frac{1}{2}};$      $\bar{v}_{s+1} = \bar{v}_{s+\frac{1}{2}} - \frac{\theta}{2}\nabla f(\bar{x}_{s+1});$
12:    **end for**
        % Metropolis filter
13:    Compute the acceptance ratio $\alpha_k = \min\left(1, \frac{\exp(-H(\bar{x}_{S_k}, \bar{v}_{S_k}))}{\exp(-H(\bar{x}_0, \bar{v}_0))}\right)$.
14:    Draw $\zeta \sim \text{Uniform}[0, 1]$.
15:    **If** $\zeta < \alpha_k$ **then**
16:        $x_k \leftarrow \bar{x}_{S_k}$
17:    **Else**
18:        $x_k \leftarrow x_{k-1}$.
19: **end for**

---

of using the constant integration time, which illustrates acceleration via the proposed Chebyshev integration time.

We now provide the proof of Theorem 1.

*Proof.* (of Theorem 1) From Lemma 2, we have

$$W_2(\rho_K, \pi) \leq \max_{j \in [d]} \left| \Pi_{k=1}^K \cos\left(\sqrt{2\lambda_j}\eta_k^{(K)}\right) \right| \cdot W_2(\rho_0, \pi). \tag{12}$$

We can upper-bound the cosine product of any $j \in [d]$ as,

$$\left| \Pi_{k=1}^K \cos\left(\sqrt{2\lambda_j}\eta_k^{(K)}\right) \right| \stackrel{(a)}{=} \left| \Pi_{k=1}^K \cos\left(\frac{\pi}{2}\sqrt{\frac{\lambda_j}{r_{\sigma(k)}^{(K)}}}\right) \right| \stackrel{(b)}{\leq} \left| \bar{\Phi}_K(\lambda_j) \right| \stackrel{(c)}{\leq} 2\left(1 - 2\frac{\sqrt{m}}{\sqrt{L}+\sqrt{m}}\right)^K, \tag{13}$$

where (a) is due to the use of Chebyshev integration time (10), (b) is by Lemma 4, and (c) is by Lemma 3. Combining (12) and (13) leads to the result. $\square$

**HMC with Chebyshev Integration Time for General Distributions**   To sample from general strongly log-concave distributions, we propose Algorithm 2, which adopts the Verlet integrator (a.k.a. the leapfrog integrator) to simulate the Hamiltonian flow $\text{HMC}_\eta(\cdot, \xi)$ and uses Metropolis filter to correct the bias. It is noted that the number of leapfrog steps $S_k$ in each iteration $k$ is equal to the integration time $\eta_k^{(K)}$ divided by the step size $\theta$ used in the leapfrog steps. More precisely, we have $S_k = \lfloor \frac{\eta_k^{(K)}}{\theta} \rfloor$ in iteration $k$ of HMC.

## 4   EXPERIMENTS

We now evaluate HMC with the proposed Chebyshev integration time (Algorithm 2) and HMC with the constant integration time (Algorithm 2 with line 7 replaced by the constant integration time (5)) in several tasks. For all the tasks in the experiments, the total number of iterations of HMCs is set to be $K = 10,000$, and hence we collect $K = 10,000$ samples along the trajectory. For the step size $\theta$ in the leapfrog steps, we let $\theta \in \{0.001, 0.005, 0.01, 0.05\}$. To evaluate the methods, we

Table 1: Ideal HMC with $K = 10,000$ iterations for sampling from a Gaussian $\mathcal{N}(\mu, \Sigma)$, where $\mu = \begin{bmatrix} 0 \\ 0 \end{bmatrix}$ and $\Sigma = \begin{bmatrix} 1 & 0 \\ 0 & 100 \end{bmatrix}$. Here, Cheby. (W/) is ideal HMC with a arbitrary permutation of the Chebyshev integration time, while Cheby. (W/O) is ideal HMC without a permutation; and Const. refers to using the constant integration time (5).

| Method | Mean ESS | Min ESS |
|---|---|---|
| Cheby. (W/) | $10399.00811 \pm 347.25021$ | $7172.50338 \pm 257.21244$ |
| Cheby. (W/O) | $10197.09964 \pm 276.94894$ | $7043.55293 \pm 284.78037$ |
| Const. | $7692.00382 \pm 207.19628$ | $5533.26519 \pm 213.31943$ |

compute effective sample size (ESS), which is a common performance metric of HMCs (Girolami & Calderhead, 2011; Brofos & Lederman, 2021; Hirt et al., 2021; Riou-Durand & Vogrinc, 2022; Hoffman et al., 2021; Hoffman & Gelman, 2014; Steeg & Galstyan, 2021), by using the toolkit ArViz (Kumar et al., 2019). The ESS of a sequence of $N$ dependent samples is computed based on the autocorrelations within the sequence at different lags: $\text{ESS} := N/(1 + 2\sum_k \gamma(k))$, where $\gamma(k)$ is an estimate of the autocorrelation at lag $k$. We consider 4 metrics, which are (1) **Mean ESS:** the average of ESS of all variables. That is, ESS is computed for each variable/dimension, and Mean ESS is the average of them. (2) **Min ESS:** the lowest value of ESS among the ESSs of all variables; (3) **Mean ESS/Sec.:** Mean ESS normalized by the CPU time in seconds; (4) **Min ESS/Sec.:** Minimum ESS normalized by the CPU time in seconds. In the following tables, we denote "Cheby." as our proposed method, and "Const." as HMC with the the constant integration time (Vishnoi, 2021; Chen & Vempala, 2019). Each of the configurations is repeated 10 times, and we report the average and the standard deviation of the results. We also report the acceptance rate of the Metropolis filter (Acc. Prob) on the tables. Our implementation of the experiments is done by modifying a publicly available code of HMCs by Brofos & Lederman (2021). Code for our experiments can be found in the supplementary.

### 4.1 IDEAL HMC FLOW FOR SAMPLING FROM A GUSSIAN WITH A DIAGONAL COVARIANCE

Before evaluating the empirical performance of Algorithm 2 in the following subsections, here we discuss and compare the use of a arbitrary permutation of the Chebyshev integration time and that without permutation (as well as that of using a constant integration time). We simulate ideal HMC for sampling from a Gaussian $\mathcal{N}(\mu, \Sigma)$, where $\mu = \begin{bmatrix} 0 \\ 0 \end{bmatrix}$ and $\Sigma = \begin{bmatrix} 1 & 0 \\ 0 & 100 \end{bmatrix}$. It is noted that ideal HMC flow for this case has a closed-form solution as (4) shows. The result are reported on Table 1.

From the table, the use of a Chebyshev integration time allows to obtain a larger ESS than that from using a constant integration time, and a arbitrary permutation helps get a better result. An explanation is that the ESS is a quantity that is computed along the trajectory of a chain, and therefore a permutation of the integration time could make a difference. We remark that the observation here (a arbitrary permutation of time generates a larger ESS) does not contradict to Theorem 1, since Theorem 1 is about the guarantee in $W_2$ distance at the last iteration $K$.

### 4.2 SAMPLING FROM A GAUSSIAN

We sample $\mathcal{N}(\mu, \Sigma)$, where $\mu = \begin{bmatrix} 0 \\ 1 \end{bmatrix}$ and $\Sigma = \begin{bmatrix} 1 & 0.5 \\ 0.5 & 100 \end{bmatrix}$. Therefore, the strong convexity constant $m$ is approximately $0.01$ and the smoothness constant $L$ is approximately $1$. Table 2 shows the results. HMC with Chebyshev integration time consistently outperforms that of using the constant integration time in terms of all the metrics: Mean ESS, Min ESS, Mean ESS/Sec, and Min ESS/Sec.

We also plot two quantities throughout the iterations of HMCs on Figure 2. Specifically, Sub-figure (a) on Figure 2 plots the size of the difference between the targeted covariance $\Sigma$ and an estimated covariance $\hat{\Sigma}_k$ at each iteration $k$ of HMC, where $\hat{\Sigma}_k$ is the sample covariance of $10,000$ samples collected from a number of $10,000$ HMC chains at their $k_{\text{th}}$ iteration. Sub-figure (b) plots a discrete TV distance that is computed as follows. We use a built-in function of Numpy to sample $10,000$ samples from the target distribution, while we also have $10,000$ samples collected from a number

Table 2: Sampling from a Gaussian distribution. We report 4 metrics regarding ESS (the higher the better), please see the main text for their definitions.

| Step Size | Method | Mean ESS | Min ESS | Mean ESS/Sec. | Min. ESS/Sec. | Acc. Prob |
|---|---|---|---|---|---|---|
| 0.001 | Cheby. | $5187.28 \pm 261.13$ | $307.09 \pm 21.92$ | $20.28 \pm 1.74$ | $1.20 \pm 0.11$ | $1.00 \pm 0.00$ |
| 0.001 | Const. | $1912.76 \pm 72.10$ | $39.87 \pm 13.77$ | $15.87 \pm 0.89$ | $0.33 \pm 0.11$ | $1.00 \pm 0.00$ |
| 0.005 | Cheby. | $5146.71 \pm 257.65$ | $304.126 \pm 19.09$ | $97.84 \pm 9.23$ | $5.79 \pm 0.68$ | $1.00 \pm 0.00$ |
| 0.005 | Const. | $1926.71 \pm 136.53$ | $32.83 \pm 9.57$ | $80.31 \pm 4.39$ | $1.37 \pm 0.39$ | $1.00 \pm 0.00$ |
| 0.01 | Cheby. | $5127.90 \pm 211.46$ | $279.59 \pm 38.09$ | $184.26 \pm 20.99$ | $10.01 \pm 1.52$ | $1.00 \pm 0.00$ |
| 0.01 | Const. | $1832.87 \pm 77.47$ | $35.71 \pm 11.74$ | $147.53 \pm 12.59$ | $2.85 \pm 0.95$ | $1.00 \pm 0.00$ |
| 0.05 | Cheby. | $5133.67 \pm 195.07$ | $316.87 \pm 36.27$ | $871.72 \pm 88.73$ | $53.54 \pm 6.22$ | $0.99 \pm 0.00$ |
| 0.05 | Const. | $1849.15 \pm 92.75$ | $34.98 \pm 14.70$ | $615.73 \pm 30.16$ | $11.70 \pm 5.07$ | $0.99 \pm 0.00$ |
| 0.1 | Cheby. | $4948.46 \pm 144.03$ | $281.66 \pm 44.79$ | $1492.96 \pm 166.21$ | $84.39 \pm 13.04$ | $0.99 \pm 0.00$ |
| 0.1 | Const. | $1852.79 \pm 132.95$ | $38.17 \pm 16.35$ | $1035.54 \pm 82.34$ | $21.44 \pm 9.51$ | $0.99 \pm 0.00$ |

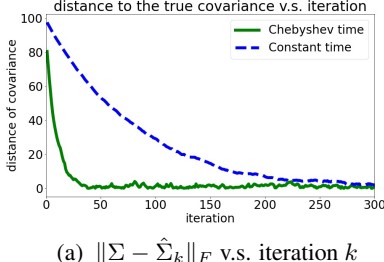

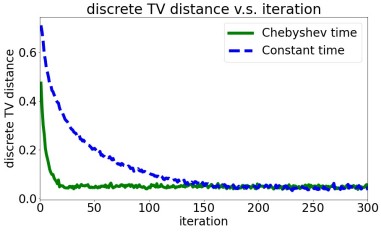

(a) $\|\Sigma - \hat{\Sigma}_k\|_F$ v.s. iteration $k$     (b) discrete $\mathrm{TV}(\hat{\pi}, \hat{\rho}_k)$ v.s. iteration $k$

Figure 2: Sampling from a Gaussian distribution. Both lines correspond to HMCs with the same step size $h = 0.05$ used in the leapfrog steps (but with different schemes of the integration time). Please see the main text for the precise definitions of the quantities and the details of how we compute them.

of $10,000$ HMC chains at each iteration $k$. Using these two sets of samples, we construct two histograms with 30 number of bins for each dimension, we denote them as $\hat{\pi}$ and $\hat{\rho}_k$. The discrete $\mathrm{TV}(\hat{\pi}, \hat{\rho}_k)$ at iteration $k$ is 0.5 times the sum of the absolute value of the difference between the number of counts of all the pairs of the bins divided by $10,000$, which serves as a surrogate of the Wasserstein-2 distance between the true target $\pi$ and $\rho_k$ from HMC, since computing or estimating the true Wasserstein distance is challenging.

### 4.3 SAMPLING FROM A MIXTURE OF TWO GAUSSIANS

For a vector $a \in \mathbb{R}^d$ and a positive definite matrix $\Sigma \in \mathbb{R}^{d \times d}$, we consider sampling from a mixture of two Gaussians $\mathcal{N}(a, \Sigma)$ and $\mathcal{N}(-a, \Sigma)$ with equal weights. Denote $b := \Sigma^{-1}a$ and $\Lambda := \Sigma^{-1}$. The potential is $f(x) = \frac{1}{2}\|x - a\|_\Lambda^2 - \log(1 + \exp(-2x^\top b))$, and its gradient is $\nabla f(x) = \Lambda x - b + 2b(1 + \exp(-2x^\top b))^{-1}$. For each dimension $i \in [d]$, we set $a[i] = \frac{\sqrt{i}}{2d}$ and set the covariance $\Sigma = \mathrm{diag}_{1 \leq i < d}(\frac{i}{d})$. The potential is strongly convex if $a^\top \Sigma^{-1} a < 1$, see e.g., Riou-Durand & Vogrinc (2022). We set $d = 10$ in the experiment, and simply use the smallest and the largest eigenvalue of $\Lambda$ to approximate the strong convexity constant $m$ and the smoothness constant $L$ of the potential, which are $\hat{m} = 1$ and $\hat{L} = 10$ in this case. Table 3 shows that the proposed method generates a larger effective sample size than the baseline.

### 4.4 BAYESIAN LOGISTIC REGRESSION

We also consider Bayesian logistic regression to evaluate the methods. Given an observation $(z_i, y_i)$, where $z_i \in \mathbb{R}^d$ and $y_i \in \{0, 1\}$, the likelihood function is modeled as $p(y_i|z_i, w) = \frac{1}{1 + \exp(-y_i z_i^\top w)}$. Moreover, the prior on the model parameter $w$ is assumed to follow a Gaussian distribution, $p(w) = N(0, \alpha^{-1}I_d)$, where $\alpha > 0$ is a parameter. The goal is to sample $w \in \mathbb{R}^d$ from the posterior, $p(w|\{z_i, y_i\}_{i=1}^n) = p(w)\Pi_{i=1}^n p(y_i|z_i, w)$, where $n$ is the number of data points in a dataset. The potential function $f(w)$ can be written as

$$f(w) = \sum_{i=1}^n f_i(w), \text{ where } f_i(w) = \log\left(1 + \exp(-y_i w^\top z_i)\right) + \alpha \frac{\|w\|^2}{2n}. \tag{14}$$

Table 3: Sampling from a mixture of two Gaussians

| Step Size | Method | Mean ESS | Min ESS | Mean ESS/Sec. | Min. ESS/Sec. | Acc. Prob |
|---|---|---|---|---|---|---|
| 0.001 | Cheby. | $2439.86 \pm 71.83$ | $815.20 \pm 83.82$ | $22.68 \pm 0.93$ | $7.57 \pm 0.81$ | $0.89 \pm 0.00$ |
| 0.001 | Const. | $845.44 \pm 31.42$ | $261.14 \pm 34.34$ | $12.90 \pm 0.52$ | $3.98 \pm 0.53$ | $0.91 \pm 0.00$ |
| 0.005 | Cheby. | $2399.50 \pm 100.12$ | $784.06 \pm 82.07$ | $105.97 \pm 8.78$ | $34.58 \pm 4.12$ | $0.89 \pm 0.00$ |
| 0.005 | Const. | $876.61 \pm 25.62$ | $277.72 \pm 30.62$ | $63.80 \pm 4.67$ | $20.22 \pm 2.62$ | $0.91 \pm 0.00$ |
| 0.01 | Cheby. | $2341.35 \pm 89.99$ | $794.27 \pm 48.75$ | $194.81 \pm 23.51$ | $66.30 \pm 9.89$ | $0.88 \pm 0.00$ |
| 0.01 | Const. | $860.61 \pm 20.39$ | $235.33 \pm 33.73$ | $110.62 \pm 14.09$ | $30.40 \pm 6.34$ | $0.91 \pm 0.00$ |
| 0.05 | Cheby. | $2214.19 \pm 87.27$ | $748.66 \pm 46.18$ | $761.59 \pm 68.88$ | $256.51 \pm 13.76$ | $0.89 \pm 0.00$ |
| 0.05 | Const. | $853.40 \pm 41.05$ | $265.70 \pm 37.41$ | $376.54 \pm 67.83$ | $116.45 \pm 22.23$ | $0.91 \pm 0.00$ |
| 0.1 | Cheby. | $2064.42 \pm 67.44$ | $657.45 \pm 60.44$ | $1162.67 \pm 84.19$ | $370.07 \pm 41.11$ | $0.90 \pm 0.00$ |
| 0.1 | Const. | $632.70 \pm 22.78$ | $182.88 \pm 37.10$ | $450.53 \pm 93.60$ | $132.58 \pm 43.91$ | $0.92 \pm 0.00$ |

We set $\alpha = 1$ in the experiments. We consider three datasets: Heart, Breast Cancer, and Diabetes binary classification datasets, which are all publicly available online. To approximate the strong convexity constant $m$ and the smoothness constant $L$ of the potential $f(w)$, we compute the smallest eigenvalue and the largest eigenvalue of the Hessian $\nabla^2 f(w)$ at the maximizer of the posterior, and we use them as estimates of $m$ and $L$ respectively. We apply Newton's method to approximately find the maximizer of the posterior. The experimental results are reported on Table 4 in Appendix E.1 due to the space limit, which show that our method consistently outperforms the baseline.

### 4.5 SAMPLING FROM A *hard* DISTRIBUTION

We also consider sampling from a step-size-dependent distribution $\pi(x) \propto \exp(-f_h(x))$, where the potential $f_h(\cdot)$ is $\kappa$-smooth and 1-strongly convex. The distribution is considered in Lee et al. (2021a) for showing a lower bound regarding certain Metropolized sampling methods using a constant integration time and a constant step size $h$ of the leapfrog integrator. More concretely, the potential is

$$f_h(x) := \sum_{i=1}^{d} f_i^{(h)}(x_i), \text{ where } f_i^{(h)}(x_i) = \begin{cases} \frac{1}{2}x_i^2, & i = 1 \\ \frac{\kappa}{3}x_i^2 - \frac{\kappa h}{3}\cos\left(\frac{x_i}{\sqrt{h}}\right), & 2 \le i \le d. \end{cases} \quad (15)$$

In the experiment, we set $\kappa = 50$ and $d = 10$. The results are reported on Table 5 in Appendix E.2. The scheme of the Chebyshev integration time is still better than the constant integration time for this task.

## 5 DISCUSSION AND OUTLOOK

The Chebyshev integration time shows promising empirical results for sampling from a various of strongly log-concave distributions. On the other hand, the theoretical guarantee of acceleration that we provide in this work is only for strongly convex quadratic potentials. Therefore, a direction left open by our work is establishing some provable acceleration guarantees for general strongly log-concave distributions. However, unlike quadratic potentials, the output (position, velocity) of a HMC flow does not have a closed-form solution in general, which makes the analysis much more challenging. A starting point might be improving the analysis of Chen & Vempala (2019), where a contraction bound of two HMC chains under a small integration time $\eta = O(\frac{1}{\sqrt{L}})$ is shown. Since the scheme of the Chebyshev integration time requires a large integration time $\eta = \Theta\left(\frac{1}{\sqrt{m}}\right)$ at some iterations of HMC, a natural question is whether a variant of the result of Chen & Vempala (2019) can be extended to a large integration time $\eta = \Theta\left(\frac{1}{\sqrt{m}}\right)$. We state as an open question: can ideal HMC with a scheme of time-varying integration time achieve an accelerated rate $O(\sqrt{\kappa}\log\frac{1}{\epsilon})$ for general smooth strongly log-concave distributions?

The topic of accelerating HMC with provable guarantees is underexplored, and we hope our work can facilitate the progress in this field. After the preprint of this work was available on arXiv, Jiang (2022) proposes a randomized integration time with partial velocity refreshment and provably shows that ideal HMC with the proposed machinery has the accelerated rate for sampling from a Gaussian distribution. Exploring any connections between the scheme of Jiang (2022) and ours can be an interesting direction.

ACKNOWLEDGMENTS

We thank the reviewers for constructive feedback, which helps improve the presentation of this paper.

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

## A  A CONNECTION BETWEEN OPTIMIZATION AND SAMPLING

To provide an intuition of why the technique of Chebyshev polynomials can help accelerate HMC for the case of the strongly convex quadratic potentials, we would like to describe the work of gradient descent with the Chebyshev step sizes Agarwal et al. (2021) in more detail, because we are going to draw a connection between optimization and sampling to showcase the intuition. Agarwal et al. (2021) provably show that gradient descent with a scheme of step sizes based on the Chebyshev Polynomials has an accelerated rate for minimizing strongly convex quadratic functions compared to GD with a constant step size, and their experiments show some promising results for minimizing smooth strongly convex functions beyond quadratics via the proposed scheme of step sizes. More precisely, define $f(w) = \frac{1}{2}w^\top A w$, where $A \in \mathbb{R}^{d \times d}$ is a positive definite matrix which has eigenvalues $L := \lambda_1 \geq \lambda_2 \geq \cdots \geq \lambda_d =: m$. Agarwal et al. (2021) consider applying gradient descent

$$w_{k+1} = w_k - \eta_k \nabla f(w_k)$$

to minimize $f(\cdot)$, where $\eta_k$ is the step size of gradient descent at iteration $k$. Let $w$ be the unique global minimizer of $f(\cdot)$. It is easy to show that the dynamic of the distance evolves as

$$w_{k+1} - w_* = (I_d - \eta_k A)(I_d - \eta_{k-1} A) \cdots (I_d - \eta_1 A)(w_1 - w_*).$$

Hence, the size of the distance to $w_*$ at iteration $K + 1$ is bounded by

$$\|w_{K+1} - w_*\| \leq \max_{j \in [d]} |\prod_{k=1}^{K}(1 - \eta_k \lambda_j)| \|w_1 - w_*\|.$$

This shows that the convergence rate of GD is governed by $\max_{j \in [d]} |\prod_{k=1}^{K}(1 - \eta_k \lambda_j)|$. By setting $\eta_k$ as the inverse of the Chebyshev root $r_k^{(K)}$ or any permuted root $r_{\sigma(k)}^{(K)}$ (see (8) for the definition), the polynomial $\prod_{k=1}^{K}(1 - \eta_k \lambda)$ is actually the $K$-degree scale-and-shifted polynomial, i.e., $\prod_{k=1}^{K}(1 - \eta_k \lambda) = \prod_{k=1}^{K}\left(1 - \frac{\lambda}{r_{\sigma(k)}^{(K)}}\right) = \bar{\Phi}_k(\lambda)$ (see (7) for the definition). It is well-known in the literature of optimization and numerical linear algebra that the $K$-degree scale-and-shifted polynomial satisfies

$$\max_{\lambda \in [m,L]} |\bar{\Phi}_K(\lambda)| \leq 2\left(1 - 2\frac{\sqrt{m}}{\sqrt{L} + \sqrt{m}}\right)^K = O\left(\left(1 - \Theta\left(\sqrt{\frac{m}{L}}\right)\right)^K\right),$$

which is restated in Lemma 3 and its proof is replicated in Appendix B of our paper for the reader's convenience. Applying this result, one gets a simple proof of the accelerated linear rate of GD with the proposed scheme of step sizes for minimizing quadratic functions. A nice blog article by Pedregosa (2021) explains this in detail.

Now we are ready to highlight its connection with HMC. In Lemma 1 of the paper, we restate a known result in HMC literature, where its proof is also replicated in Appendix B for the reader's convenience. The lemma indicates that the convergence rate of HMC is governed by $\max_{j \in [d]} | \prod_{k=1}^{K} \cos(\sqrt{2\lambda_j}\eta_k^{(K)})|$. By way of comparison to that of GD for minimizing quadratic functions, i.e., $\max_{j \in [d]} | \prod_{k=1}^{K}(1 - \eta_k\lambda_j)|$, it appears that they share some similarity, which made us wonder if we could bound the former by the latter. We show in Lemma 4 that $\cos(\frac{\pi}{2}\sqrt{x}) \leq 1 - x$, which holds for all $x \geq 0$, and consequently,

$$|P_K^{\text{Cos}}(\lambda)| := \left| \prod_{k=1}^{K} \cos\left( \frac{\pi}{2} \sqrt{\frac{\lambda}{r_{\sigma(k)}^{(K)}}} \right) \right| \leq \left| \prod_{k=1}^{K} \left( 1 - \frac{\lambda}{r_{\sigma(k)}^{(K)}} \right) \right| = \left| \bar{\Phi}_K(\lambda) \right|,$$

The key lemma above implies that if we set the integration time as $\eta_k^{(K)} = \frac{\pi}{2} \frac{1}{\sqrt{2r_{\sigma(k)}^{(K)}}}$, then we get acceleration of HMC.

## B  PROOFS OF LEMMAS IN SECTION 2

We restate the lemmas for the reader's convenience.

**Lemma 1.** (Vishnoi, 2021) *Let* $x_0, y_0 \in \mathbb{R}^d$. *Consider the following coupling:* $(x_t, v_t) = \text{HMC}_t(x_0, \xi)$ *and* $(y_t, u_t) = \text{HMC}_t(y_0, \xi)$ *for some* $\xi \in \mathbb{R}^d$. *Then for all* $t \geq 0$ *and for all* $j \in [d]$, *it holds that*

$$x_t[j] - y_t[j] = \cos\left( \sqrt{2\lambda_j}t \right) \times (x_0[j] - y_0[j]).$$

*Proof.* Given $(x_t, v_t) := \text{HMC}_t(x_0, \xi)$ and $(y_t, u_t) := \text{HMC}_t(y_0, \xi)$, we have $\frac{dv_t}{dt} - \frac{du_t}{dt} = -\nabla f(x_t) + \nabla f(y_t) = 2\Lambda(y_t - x_t)$. Therefore, we have $\frac{d^2(x_t[j] - y_t[j])}{dt^2} = -2\lambda_j(x_t[j] - y_t[j])$, for all $j \in [d]$. Because of the initial condition $\frac{dx_0[j]}{dt} = \frac{dy_0[j]}{dt} = \xi[j]$, the differential equation implies that $x_t[j] - y_t[j] = \cos\left( \sqrt{2\lambda_j}t \right) \times (x_0[j] - y_0[j])$.

It is noted that the result also follows directly from the explicit solution (4). $\square$

**Lemma 2.** (Vishnoi, 2021) *Let* $\pi \propto \exp(-f) = \mathcal{N}(0, \Lambda^{-1})$ *be the target distribution, where* $f(x)$ *is defined on* (3). *Let* $\rho_K$ *be the distribution of* $x_K$ *generated by Algorithm 1 at the final iteration* $K$. *Then for any* $\rho_0$ *and any* $K \geq 1$, *we have*

$$W_2(\rho_K, \pi) \leq \max_{j \in [d]} \left| \Pi_{k=1}^{K} \cos\left( \sqrt{2\lambda_j}\eta_k^{(K)} \right) \right| W_2(\rho_0, \pi).$$

*Proof.* Starting from $x_0 \sim \rho_0$, draw an initial point $y_0 \sim \pi$ such that $(x_0, y_0)$ has the optimal $W_2$-coupling between $\rho_0$ and $\pi$. Consider the following coupling at each iteration $k$: $(x_k, v_k) = \text{HMC}_{\eta_k^{(K)}}(x_{k-1}, \xi_k)$ and $(y_k, u_k) = \text{HMC}_{\eta_k^{(K)}}(y_{k-1}, \xi_k)$ where $\xi_k \sim \mathcal{N}(0, I)$ is an independent Gaussian. We collect $\{x_k\}_{k=1}^{K}$ and $\{y_k\}_{k=1}^{K}$ from Algorithm 1. We know each $y_k \sim \pi$, since $\pi$ is a

stationary distribution of the HMC Markov chain. Then by Lemma 1 we have

$$
\begin{aligned}
W_2^2(\rho_K, \pi) &\leq \mathbb{E}[\|x_K - y_K\|^2] \\
&= \mathbb{E}[\textstyle\sum_{j\in[d]}(x_K[j] - y_K[j])^2] \\
&= \mathbb{E}[\textstyle\sum_{j\in[d]} \left(\Pi_{k=1}^K \cos\left(\sqrt{2\lambda_j}\eta_k^{(K)}\right) \times (x_0[j] - y_0[j])\right)^2] \\
&\leq \left(\max_{j\in[d]}\left(\Pi_{k=1}^K \cos\left(\sqrt{2\lambda_j}\eta_k^{(K)}\right)\right)^2\right)\mathbb{E}[\textstyle\sum_{j\in[d]}(x_0[j] - y_0[j])^2] \\
&= \left(\max_{j\in[d]}\left(\Pi_{k=1}^K \cos\left(\sqrt{2\lambda_j}\eta_k^{(K)}\right)\right)^2\right) W_2^2(\rho_0, \pi),
\end{aligned}
\tag{16}
$$

Taking the square root on both sides leads to the result. $\qquad\square$

**Lemma 3.** *(e.g., Section 2.3 in d'Aspremont et al. (2021)) For any positive integer $K$, we have*

$$
\max_{\lambda\in[m,L]}\left|\bar{\Phi}_K(\lambda)\right| \leq 2\left(1 - 2\frac{\sqrt{m}}{\sqrt{L}+\sqrt{m}}\right)^K = O\left(\left(1 - \Theta\left(\sqrt{\tfrac{m}{L}}\right)\right)^K\right).
\tag{17}
$$

*Proof.* Observe that the numerator of $\bar{\Phi}_K(\lambda) = \frac{\Phi_K(h(\lambda))}{\Phi_K(h(0))}$ satisfies $|\Phi_K(h(\lambda))| \leq 1$, since $h(\lambda) \in [-1, 1]$ for $\lambda \in [m, L]$ and that the Chebyshev polynomial satisfies $|\Phi_K(\cdot)| \leq 1$ when its argument is in $[-1, 1]$ by the definition. It remains to bound the denominator, which is $\Phi_K(h(0)) = \cosh\left(K \operatorname{arccosh}\left(\frac{L+m}{L-m}\right)\right)$. Since

$$
\operatorname{arccosh}\left(\tfrac{L+m}{L-m}\right) = \log\left(\tfrac{L+m}{L-m} + \sqrt{\left(\tfrac{L+m}{L-m}\right)^2 - 1}\right) = \log(\theta), \text{ where } \theta := \tfrac{\sqrt{L}+\sqrt{m}}{\sqrt{L}-\sqrt{m}},
$$

we have

$$
\Phi_K(h(0)) = \cosh\left(K \operatorname{arccosh}\left(\tfrac{L+m}{L-m}\right)\right) = \tfrac{\exp(K\log(\theta)) + \exp(-K\log(\theta))}{2} = \tfrac{\theta^K + \theta^{-K}}{2} \geq \tfrac{\theta^K}{2}.
$$

Combing the above inequalities, we obtain the desired result:

$$
\begin{aligned}
\max_{\lambda\in[m,L]}\left|\bar{\Phi}_K(\lambda)\right| = \max_{\lambda\in[m,L]}\left|\frac{\Phi_K(h(\lambda))}{\Phi_K(h(0))}\right| &\leq \frac{2}{\theta^K} = 2\left(1 - 2\frac{\sqrt{m}}{\sqrt{L}+\sqrt{m}}\right)^K \\
&= O\left(\left(1 - \Theta\left(\sqrt{\tfrac{m}{L}}\right)\right)^K\right).
\end{aligned}
$$

$\qquad\square$

## C  PROOF OF LEMMA 4

**Lemma 4.** *Denote $|P_K^{\mathrm{Cos}}(\lambda)| := \left|\Pi_{k=1}^K \cos\left(\frac{\pi}{2}\sqrt{\frac{\lambda}{r_{\sigma(k)}^{(K)}}}\right)\right|$. Suppose $\lambda \in [m, L]$. Then, we have for any positive integer $K$,*

$$
|P_K^{\mathrm{Cos}}(\lambda)| \leq \left|\bar{\Phi}_K(\lambda)\right|.
\tag{18}
$$

*Proof.* We use the fact that the $K$-degree scaled-and-shifted Chebyshev Polynomial can be written as,

$$
\bar{\Phi}_K(\lambda) = \Pi_{k=1}^K\left(1 - \frac{\lambda}{r_{\sigma(k)}^{(K)}}\right),
\tag{19}
$$

for any permutation $\sigma(\cdot)$, since $\{r_{\sigma(k)}^{(K)}\}$ are its roots and $\bar{\Phi}_K(0) = 1$. So inequality (18) is equivalent to

$$\left| \Pi_{k=1}^K \cos\left( \frac{\pi}{2} \sqrt{\frac{\lambda}{r_{\sigma(k)}^{(K)}}} \right) \right| \leq \left| \Pi_{k=1}^K \left( 1 - \frac{\lambda}{r_{\sigma(k)}^{(K)}} \right) \right|. \tag{20}$$

To show (20), let us analyze the mapping $\psi(x) := \frac{\cos\left(\frac{\pi}{2}\sqrt{x}\right)}{1-x}$ for $x \geq 0$, $x \neq 1$, with $\psi(1) = \frac{\pi}{4}$ by continuity, and show that $\max_{x:x\geq 0} |\psi(x)| \leq 1$, as (20) would be immediate. We have $\psi'(x) = -\frac{\pi}{4\sqrt{x}} \frac{1}{1-x} \sin(\frac{\pi}{2}\sqrt{x}) + \cos(\frac{\pi}{2}\sqrt{x}) \frac{1}{(1-x)^2}$. Hence, $\psi'(x) = 0$ when

$$\tan(\tfrac{\pi}{2}\sqrt{x}) = \frac{4\sqrt{x}}{\pi(1-x)}. \tag{21}$$

Denote an extreme point of $\psi(x)$ as $\hat{x}$, which satisfies (21). Then, using (21), we have $|\psi(\hat{x})| = \left| \frac{\cos\left(\frac{\pi}{2}\sqrt{\hat{x}}\right)}{1-\hat{x}} \right| = \left| \frac{\pi}{\sqrt{16\hat{x}+\pi^2(1-\hat{x})^2}} \right|$, where we used $\cos(\frac{\pi}{2}\sqrt{\hat{x}}) = \frac{\pi(1-\hat{x})}{\sqrt{16\hat{x}+\pi^2(1-\hat{x})^2}}$ or $\frac{-\pi(1-\hat{x})}{\sqrt{16\hat{x}+\pi^2(1-\hat{x})^2}}$. The denominator $\sqrt{16\hat{x}+\pi^2(1-\hat{x})^2}$ has the smallest value at $\hat{x} = 0$, which means that the largest value of $|\psi(x)|$ happens at $x = 0$, which is 1. The proof is now completed.

$\square$

# D  A COMPARISON OF THE TOTAL INTEGRATION TIME (JIANG, 2022)

Since the Chebyshev integration time are set to be some large values at some steps of HMC, it is natural to ask if the number of steps to get an $\epsilon$ 2-Wasserstein distance is a fair metric. In this section, we consider the total integration time $\sum_{k=1}^K \eta_k^{(K)}$ to get an $\epsilon$ distance as another metric for the convergence. It is noted that the comparison between HMC with our integration time and HMC with the best constant integration time has been conducted by Jiang (2022), and our previous version did not have such a comparison. Below, we reproduce the comparision of Jiang (2022).

Recall the number of iterations to get an $\epsilon$ 2-Wasserstein distance to the target distribution is $K = O\left(\sqrt{\kappa}\log\left(\frac{1}{\epsilon}\right)\right)$ of HMC with the Chebyshev integration time (Theorem 1 in the paper). The average of the integration time is

$$\frac{1}{K}\sum_{k=1}^K \eta_k^{(K)} = \frac{1}{K}\sum_{k=1}^K \frac{\pi}{2\sqrt{2}} \frac{1}{\sqrt{r_{\sigma(k)}^{(K)}}} = \frac{1}{K}\sum_{k=1}^K \frac{\pi}{2\sqrt{2}} \frac{1}{\sqrt{r_k^{(K)}}},$$

where we recall that a permutation $\sigma(\cdot)$ does not affect the average.

Then, if $K$ is even, we can rewrite the averaged integration time as

$$\frac{1}{K}\sum_{k=1}^K \eta_k^{(K)} = \frac{1}{K}\frac{\pi}{2\sqrt{2}}\sum_{k=1}^{K/2}\left( \frac{1}{\sqrt{r_k^{(K)}}} + \frac{1}{\sqrt{r_{K+1-k}^{(K)}}} \right).$$

Otherwise, $K$ is odd, and we can rewrite the averaged integration time as

$$\frac{1}{K}\sum_{k=1}^K \eta_k^{(K)} = \frac{1}{K}\frac{\pi}{2\sqrt{2}}\left( \frac{1}{\sqrt{r_{(K+1)/2}^{(K)}}} + \sum_{k=1}^{(K-1)/2}\left( \frac{1}{\sqrt{r_k^{(K)}}} + \frac{1}{\sqrt{r_{K+1-k}^{(K)}}} \right) \right).$$

We will show

$$\frac{1}{\sqrt{r_k^{(K)}}} + \frac{1}{\sqrt{r_{K+1-k}^{(K)}}} \leq \frac{1}{\sqrt{r_{\lfloor K/2 \rfloor}^{(K)}}} + \frac{1}{\sqrt{r_{K-\lfloor K/2 \rfloor+1}^{(K)}}},$$

for any $k = \{1, 2, \ldots, \lfloor \frac{K}{2} \rfloor\}$ soon. Given this, we can further upper-bound the averaged integration time as

$$\frac{1}{K}\sum_{k=1}^K \eta_k^{(K)} \leq \frac{\pi}{4\sqrt{2}}\left( \frac{1}{\sqrt{r_{\lfloor K/2 \rfloor}^{(K)}}} + \frac{1}{\sqrt{r_{K-\lfloor K/2 \rfloor+1}^{(K)}}} \right),$$

when $K$ is even; when $K$ is odd, we can upper-bound the averaged integration time as

$$\frac{1}{K} \sum_{k=1}^{K} \eta_k^{(K)} \leq \frac{1}{K} \frac{\pi}{2\sqrt{2}} \left( \frac{1}{\sqrt{r_{(K+1)/2}^{(K)}}} + \frac{K-1}{2} \left( \frac{1}{\sqrt{r_{\lfloor K/2 \rfloor}^{(K)}}} + \frac{1}{\sqrt{r_{K-\lfloor K/2 \rfloor+1}^{(K)}}} \right) \right).$$

Using the definition of the Chebyshev root, we have

$$r_{\lfloor K/2 \rfloor}^{(K)} = \frac{L+m}{2} - \frac{L-m}{2}\cos\left( \frac{\left( \lfloor \frac{K}{2} \rfloor - \frac{1}{2}\right)\pi}{K} \right) \approx \frac{L+m}{2},$$

where the approximation is because $\frac{\left(\lfloor \frac{K}{2} \rfloor - \frac{1}{2}\right)\pi}{K} \approx \frac{\pi}{2}$ when $K$ is large, and hence $\cos\left( \frac{\left(\lfloor \frac{K}{2} \rfloor - \frac{1}{2}\right)\pi}{K} \right) \approx$ 0. Similarly, we can approximate

$$r_{K-\lfloor K/2 \rfloor+1}^{(K)} = \frac{L+m}{2} - \frac{L-m}{2}\cos\left( \frac{\left( K - \lfloor K/2 \rfloor + 1 - \frac{1}{2}\right)\pi}{K} \right) \approx \frac{L+m}{2}$$

as $\frac{\left( K - \lfloor K/2 \rfloor + 1 - \frac{1}{2}\right)\pi}{K} \approx \frac{\pi}{2}$ when $K$ is large. Also, we can approximate $r_{(K+1)/2}^{(K)} \approx \frac{L+m}{2}$ when $K$ is odd and large for the same reason.

Combining the above, the total integration time of HMC with the Chebyshev scheme can be approximated as

$$\text{number of iterations} \times \text{average integration time}$$

$$= \sqrt{\kappa} \log\left( \frac{1}{\epsilon} \right) \times \frac{1}{K} \sum_{k=1}^{K} \eta_k^{(K)} \approx \sqrt{\kappa} \log\left( \frac{1}{\epsilon} \right) \times \frac{\pi}{2} \frac{1}{\sqrt{L+m}}.$$

When $\kappa := \frac{L}{m}$ is large, the total integration time becomes

$$\sqrt{\kappa} \log\left( \frac{1}{\epsilon} \right) \times \frac{\pi}{2} \frac{1}{\sqrt{L+m}} = \Theta\left( \frac{1}{\sqrt{m}} \log\left( \frac{1}{\epsilon} \right) \right). \tag{22}$$

Now let us switch to analyzing HMC with the best constant integration time $\eta = \Theta\left( \frac{1}{\sqrt{L}} \right)$ (see e.g., (5), Vishnoi (2021)), which has the non-accelerated rate. Specifically, it needs $K = O\left( \kappa \log\left( \frac{1}{\epsilon} \right) \right)$ iterations to converge to the target distribution. Hence, the total integration time of HMC with the best constant integration time is

$$\text{number of iterations} \times \text{average integration time} = \kappa \log\left( \frac{1}{\epsilon} \right) \times \Theta\left( \frac{1}{\sqrt{L}} \right) = \Theta\left( \frac{\sqrt{L}}{m} \log\left( \frac{1}{\epsilon} \right) \right). \tag{23}$$

By way of comparison ((22) vs. (23)), we see that the total integration time of HMC with the proposed scheme of Chebyshev integration time reduces by a factor $\sqrt{\kappa}$, compared with HMC with the best constant integration time.

The remaining thing to show is the inequality

$$\frac{1}{\sqrt{r_k^{(K)}}} + \frac{1}{\sqrt{r_{K+1-k}^{(K)}}} \leq \frac{1}{\sqrt{r_{\lfloor K/2 \rfloor}^{(K)}}} + \frac{1}{\sqrt{r_{K+1-\lfloor K/2 \rfloor}^{(K)}}}, \tag{24}$$

for any $k = \{1, 2, \ldots, \lfloor \frac{K}{2} \rfloor\}$.

We have

$$\frac{1}{\sqrt{r_k^{(K)}}} + \frac{1}{\sqrt{r_{K+1-k}^{(K)}}}$$

$$= \sqrt{2} \times \left( \frac{1}{\sqrt{L + m - (L - m)\cos\left(\frac{\left(k - \frac{1}{2}\right)\pi}{K}\right)}} + \frac{1}{\sqrt{L + m - (L - m)\cos\left(\frac{\left(K - k + \frac{1}{2}\right)\pi}{K}\right)}} \right)$$

$$= \sqrt{2} \times \left( \frac{1}{\sqrt{L + m - (L - m)\cos\left(\frac{\left(k - \frac{1}{2}\right)\pi}{K}\right)}} + \frac{1}{\sqrt{L + m + (L - m)\cos\left(\frac{\left(k - \frac{1}{2}\right)\pi}{K}\right)}} \right).$$

$$(25)$$

Now let us define $H(k) := \left( \frac{1}{\sqrt{L + m - (L - m)\cos\left(\frac{\left(k - \frac{1}{2}\right)\pi}{K}\right)}} + \frac{1}{\sqrt{L + m + (L - m)\cos\left(\frac{\left(k - \frac{1}{2}\right)\pi}{K}\right)}} \right)$ and treat $k$ as a continuous variable.

The derivative of $H(k)$ is

$$H'(k) = \frac{\pi}{2K}(L - m)\sin\left(\frac{\left(k - \frac{1}{2}\right)\pi}{K}\right) \times$$

$$\left( \frac{1}{\left(L + m - (L - m)\cos\left(\frac{\left(k - \frac{1}{2}\right)\pi}{K}\right)\right)^{3/2}} - \frac{1}{\left(L + m + (L - m)\cos\left(\frac{\left(k - \frac{1}{2}\right)\pi}{K}\right)\right)^{3/2}} \right)$$

$$> 0. \qquad (26)$$

That is, $H'(k)$ is an increasing function of $k$ when $1 \leq k \leq \lfloor \frac{K}{2} \rfloor$, which implies that the inequality (24). Now we have completed the analysis.

# E    EXPERIMENTS

## E.1    BAYESIAN LOGISTIC REGRESSION

Table 4: Bayesian logistic regression

**HEART dataset** ($\hat{m} = 2.59$, $\hat{L} = 92.43$)

| Step Size | Method | Mean ESS | Min ESS | Mean ESS/Sec. | Min. ESS/Sec. | Acc. Prob |
|---|---|---|---|---|---|---|
| 0.001 | Cheby. | $1693.71 \pm 63.53$ | $520.43 \pm 62.24$ | $18.54 \pm 2.88$ | $5.69 \pm 1.12$ | $1.00 \pm 0.00$ |
| 0.001 | Const. | $312.18 \pm 12.65$ | $80.97 \pm 15.97$ | $6.57 \pm 0.42$ | $1.69 \pm 0.28$ | $1.00 \pm 0.00$ |
| 0.005 | Cheby. | $1664.87 \pm 43.72$ | $481.76 \pm 49.00$ | $82.90 \pm 16.51$ | $24.08 \pm 5.72$ | $0.99 \pm 0.00$ |
| 0.005 | Const. | $329.48 \pm 13.15$ | $75.78 \pm 17.30$ | $31.87 \pm 2.73$ | $7.40 \pm 2.06$ | $0.99 \pm 0.00$ |
| 0.01 | Cheby. | $1648.25 \pm 47.50$ | $508.69 \pm 49.81$ | $157.09 \pm 26.70$ | $48.45 \pm 9.64$ | $0.99 \pm 0.00$ |
| 0.01 | Const. | $307.52 \pm 8.77$ | $82.85 \pm 13.88$ | $53.89 \pm 6.37$ | $14.62 \pm 3.28$ | $0.99 \pm 0.00$ |
| 0.05 | Cheby. | $1424.21 \pm 54.03$ | $439.88 \pm 56.25$ | $458.56 \pm 51.33$ | $140.51 \pm 16.58$ | $0.98 \pm 0.00$ |
| 0.05 | Const. | $242.44 \pm 14.61$ | $56.42 \pm 17.68$ | $103.36 \pm 12.64$ | $23.90 \pm 7.40$ | $0.98 \pm 0.00$ |

**BREAST CANCER dataset** ($\hat{m} = 1.81$, $\hat{L} = 69.28$)

| Step Size | Method | Mean ESS | Min ESS | Mean ESS/Sec. | Min. ESS/Sec. | Acc. Prob |
|---|---|---|---|---|---|---|
| 0.001 | Cheby. | $1037.98 \pm 34.46$ | $575.72 \pm 41.14$ | $9.40 \pm 0.31$ | $5.21 \pm 0.31$ | $1.00 \pm 0.00$ |
| 0.001 | Const. | $174.73 \pm 13.91$ | $78.24 \pm 23.28$ | $2.59 \pm 0.29$ | $2.59 \pm 0.29$ | $1.00 \pm 0.00$ |
| 0.005 | Cheby. | $1010.49 \pm 24.15$ | $571.03 \pm 36.64$ | $43.09 \pm 1.14$ | $24.35 \pm 1.70$ | $0.99 \pm 0.00$ |
| 0.005 | Const. | $173.17 \pm 11.40$ | $79.76 \pm 13.49$ | $11.88 \pm 1.39$ | $11.88 \pm 1.39$ | $0.99 \pm 0.00$ |
| 0.01 | Cheby. | $1038.10 \pm 31.48$ | $565.54 \pm 50.51$ | $82.82 \pm 3.51$ | $45.14 \pm 4.44$ | $0.99 \pm 0.00$ |
| 0.01 | Const. | $162.64 \pm 9.43$ | $58.79 \pm 16.02$ | $18.92 \pm 2.59$ | $18.92 \pm 2.59$ | $0.99 \pm 0.00$ |
| 0.05 | Cheby. | $886.24 \pm 38.92$ | $499.54 \pm 43.99$ | $240.08 \pm 12.55$ | $135.28 \pm 12.04$ | $0.98 \pm 0.00$ |
| 0.05 | Const. | $99.48 \pm 10.10$ | $44.70 \pm 13.23$ | $33.25 \pm 6.50$ | $33.25 \pm 6.50$ | $0.98 \pm 0.00$ |

**DIABETES dataset** ($\hat{m} = 4.96$, $\hat{L} = 270.20$)

| Step Size | Method | Mean ESS | Min ESS | Mean ESS/Sec. | Min. ESS/Sec. | Acc. Prob |
|---|---|---|---|---|---|---|
| 0.001 | Cheby. | $726.08 \pm 33.92$ | $424.59 \pm 58.77$ | $11.64 \pm 0.85$ | $6.83 \pm 1.16$ | $0.99 \pm 0.00$ |
| 0.001 | Const. | $100.50 \pm 9.32$ | $41.84 \pm 19.33$ | $3.6 \pm 0.31$ | $1.50 \pm 0.68$ | $0.99 \pm 0.00$ |
| 0.005 | Cheby. | $731.46 \pm 33.04$ | $395.82 \pm 47.98$ | $54.92 \pm 5.26$ | $29.61 \pm 3.75$ | $0.99 \pm 0.00$ |
| 0.005 | Const. | $100.16 \pm 11.83$ | $44.62 \pm 20.81$ | $14.71 \pm 2.52$ | $6.67 \pm 3.37$ | $0.99 \pm 0.00$ |
| 0.01 | Cheby. | $687.74 \pm 29.31$ | $399.44 \pm 45.01$ | $93.10 \pm 6.78$ | $53.90 \pm 5.38$ | $0.98 \pm 0.00$ |
| 0.01 | Const. | $83.04 \pm 9.36$ | $36.39 \pm 12.43$ | $20.87 \pm 3.31$ | $9.09 \pm 3.25$ | $0.98 \pm 0.00$ |
| 0.05 | Cheby. | $546.80 \pm 37.40$ | $330.09 \pm 34.31$ | $206.07 \pm 17.76$ | $125.07 \pm 18.87$ | $0.96 \pm 0.00$ |
| 0.05 | Const. | $57.11 \pm 9.52$ | $23.44 \pm 9.57$ | $27.23 \pm 5.18$ | $11.02 \pm 4.34$ | $0.96 \pm 0.00$ |

## E.2    SAMPLING FROM A *hard* DISTRIBUTION

Table 5: Sampling from a distribution $\pi(x) \propto \exp(-f_h(x))$ whose potential $f_h(\cdot)$ is defined on (15).

| Step Size | Method | Mean ESS | Min ESS | Mean ESS/Sec. | Min. ESS/Sec. | Acc. Prob |
|---|---|---|---|---|---|---|
| sampling from $\pi(x) \propto \exp(-f_{0.001}(x))$ | | | | | | |
| 0.001 | Cheby. | $6222.21 \pm 88.90$ | $453.03 \pm 30.35$ | $114.74 \pm 7.59$ | $8.36 \pm 0.83$ | $1.00 \pm 0.00$ |
| 0.001 | Const. | $2098.18 \pm 46.56$ | $63.53 \pm 15.00$ | $82.31 \pm 5.29$ | $2.50 \pm 0.63$ | $1.00 \pm 0.00$ |
| sampling from $\pi(x) \propto \exp(-f_{0.005}(x))$ | | | | | | |
| 0.005 | Cheby. | $6271.43 \pm 117.71$ | $429.42 \pm 34.52$ | $545.76 \pm 26.10$ | $37.28 \pm 2.29$ | $0.99 \pm 0.00$ |
| 0.005 | Const. | $2125.36 \pm 21.87$ | $67.42 \pm 16.51$ | $361.14 \pm 5.65$ | $11.44 \pm 2.76$ | $0.99 \pm 0.00$ |
| sampling from $\pi(x) \propto \exp(-f_{0.01}(x))$ | | | | | | |
| 0.01 | Cheby. | $6523.21 \pm 95.65$ | $459.48 \pm 38.83$ | $1070.77 \pm 68.78$ | $75.61 \pm 9.79$ | $0.99 \pm 0.00$ |
| 0.01 | Const. | $2125.04 \pm 31.83$ | $69.66 \pm 20.75$ | $528.35 \pm 80.17$ | $17.19 \pm 6.34$ | $0.99 \pm 0.00$ |
| sampling from $\pi(x) \propto \exp(-f_{0.05}(x))$ | | | | | | |
| 0.05 | Cheby. | $6457.21 \pm 110.05$ | $375.97 \pm 30.64$ | $3319.51 \pm 134.92$ | $193.06 \pm 14.49$ | $0.97 \pm 0.00$ |
| 0.05 | Const. | $2796.41 \pm 56.89$ | $62.33 \pm 13.26$ | $1893.99 \pm 57.23$ | $42.22 \pm 9.05$ | $0.97 \pm 0.00$ |

