# OpenReview forum: "Accelerating Hamiltonian Monte Carlo via Chebyshev Integration Time"
_ICLR.cc/2023/Conference — ICLR 2023 poster_

### Official Review · Reviewer_XHWf · 2022-10-25

**Confidence:** 3
**Correctness:** 3
**Technical Novelty And Significance:** 3
**Empirical Novelty And Significance:** 3
**Recommendation:** 8

**Clarity, Quality, Novelty And Reproducibility:**

Clarity: This paper is well-written in terms of presenting the proposed method. However, the intuition of why this Chebyshev integration time accelerates the Ideal HMC method remains unclear.

Novelty: The proposed Chebyshev integration time is novel as it improves over the lower bound for constant integration time. Empirical evidence suggests that the same technique may also improve the efficiency of sampling for non-Gaussian tasks.

**Strength And Weaknesses:**

Strength:
This paper proposes the Chebyshev integration time which leads to a provably faster instance of the Ideal HMC method, in the Gaussian case. Empirical evidence shows that the improved rate generalizes to non-Gaussian tasks.

Weakness:
The current theoretical result only applies to the Gaussian case.


**Summary Of The Paper:**

This paper considers the technique of sampling from a Gibbs distribution via the Ideal Hamiltonian Monte-Carlo (HMC) method. Specifically, the ideal HMC method is derived by iterative running the HMC from the current spatial position, but with a resampled velocity for a certain period of integration time. The integration time plays a key role in proving the mixing time of the algorithm: It is known that even for the Gaussian case, with a constant integration time, the mixing time is proportional to $\Omega(\kappa)$. To improve this, this paper proposes the Chebyshev integration time, which are derived from the roots of the Chebyshev polynomials. It is proved that using the proposed integrating time, the resulting ideal HMC method enjoys a mixing time of order $O(\sqrt{\kappa})$, for the Gaussian case. The authors also provide empirical evidence that the improved rate generalizes to non-Gaussian tasks.

**Summary Of The Review:**

The Chebyshev integration time is novel, but the current analysis is limited to the Gaussian case. Intuition behind the success of this technique is missing.

---

> ### Author Response · Authors · 2022-11-12
> **Thanks for the positive feedback and comments**
>
> Thanks for the positive feedback and comments.
>
> Regarding the intuition behind the success of this technique, Reviewer Kxzm also made a similar comment. Here is our response to the comment.
>
> To provide an intuition of why the technique of Chebyshev polynomials can help accelerate HMC for the case of the strongly convex quadratic potentials, we would like to describe the work of gradient descent with the Chebyshev step sizes (Agarwal et al. (2021)) in more detail, because we are going to draw a connection between optimization and sampling to showcase the intuition.
> Agarwal et al. (2021) provably show that gradient descent with a scheme of step sizes based on the Chebyshev Polynomials has an accelerated rate for minimizing strongly convex quadratic functions compared to GD with a constant step size, and their experiments show some promising results for minimizing smooth strongly convex functions beyond quadratics via the proposed scheme of step sizes.
> More precisely, define $f(w)= \frac{1}{2} w^{\top} A w$, where $A \in \mathbb{R}^{{d \times d}}$ is a positive definite matrix which has eigenvalues
> $L:=\lambda_{1} \geq \lambda_{2} \geq \dots \geq \lambda_{{d}}=:m$.
> Agarwal et al.\ (2021) consider applying gradient descent
> $$ w_{k+1} = w_k - \eta_k \nabla f(w_k)$$
> to minimize $f(\cdot)$,
> where $\eta_{k}$ is the step size of gradient descent at iteration $k$.
> Let $w_{\*}$ be the unique global minimizer of $f(\cdot)$.
> It is easy to show that the dynamic of the distance evolves as
>
> $$ w_{k+1}-w_* = (I_d - \eta_k A) (I_d - \eta_{k-1} A) \cdots (I_d - \eta_{1} A) (w_1 - w_*).$$
>
> Hence, the size of the distance to $w_*$ at iteration $K+1$ is bounded by
>
> $$ \\\| w_{K+1} - w_* \\\| \leq \max_{ j \in [d] }  | \prod_{k=1}^K  ( 1 - \eta_k \lambda_j ) | \\\| w_1  - w_* \\\|.$$
>
> This shows that the convergence rate of GD is governed by
> $\max_{ j \in [d] } | \prod_{k=1}^K  ( 1 - \eta_k \lambda_j ) | $.
> By setting $\eta_{k}$ as the inverse of the Chebyshev root
> $r_{k}^{(K)}$ or any permuted root $r_{ \sigma(k) }^{(K)}$ (see eq. (8) in our paper for the definition),
> the polynomial
> $ \prod_{k=1}^K  ( 1 - \eta_k \lambda ) $ is actually
> the $K$-degree scale-and-shifted polynomial, i.e.,
> $ \prod_{k=1}^K  ( 1 - \eta_k \lambda ) =\prod_{k=1}^K \left( 1 - \frac{\lambda}{r_{\sigma(k)}^{(K)}}  \right) = \bar{{\Phi}}_{k}(\lambda)$
>
> (see eq. (7) in our paper for the definition).
> It is well-known in the literature of optimization and numerical linear algebra that
> the $K$-degree scale-and-shifted polynomial satisfies
>
> $$  \max_{\lambda \in [m,L]} \left| \bar{\Phi}_K(\lambda) \right| \leq 2 \left( 1 - 2 \frac{ \sqrt{m} }{ \sqrt{L} + \sqrt{m} } \right)^{K}
> = O\left( \left( 1 - \Theta\left( \sqrt{ \frac{m}{L} }  \right)\right)^K  \right),$$
>
> which is restated in Lemma 3 of our paper and its proof is replicated in Appendix~A of our paper for the reader's convenience. Applying this result, one gets a simple proof of the accelerated linear rate of GD with the proposed scheme of step sizes for minimizing quadratic functions. A nice blog article by Pedregosa (2021) explains this in detail.
>
> Now we are ready to highlight its connection with HMC.
> In Lemma 2 of the paper, we restate a known result in HMC literature,
> where its proof is also replicated in Appendix A of the paper for the reader's convenience.
> The lemma states that:
>
> *For sampling distributions with strongly convex quadratic potentials, the 2-Wasserstein distance
> to the target distribution $\pi$ evolves as
> $$W_{2}(\rho_K, \pi) \leq
> \max_{j \in [d]} \left| \prod_{k=1}^K \mathrm{cos}\left( \sqrt{2 \lambda_j} \eta_k^{(K)} \right) \right|  W_{2}( \rho_0,\pi),$$
> where $\rho_{K}$ is the distribution of samples generated from ideal HMC at iteration $K$ and $\\{ \lambda_{j} \\}_{{j=1}}^{d}$ are the eigenvalues of the quadratic matrix.*
>
> It is noted that the lemma holds for any scheme of integration time $\\{ \eta_k^{(K)} \\}_{k=1}^K$.
>
> The lemma indicates that the convergence rate of HMC is governed by $\max_{j \in [d]} | \prod_{k=1}^K \mathrm{cos}( \sqrt{2 \lambda_j} \eta_k^{(K)} ) |$. By way of comparison to that
> of GD for minimizing quadratic functions,
> i.e., $\max_{ j \in [d] }
> | \prod_{k=1}^K  ( 1 - \eta_k \lambda_j ) | $, it appears that they share some similarity,
> which made us wonder if we could bound the former by the latter.
> We show in Lemma 4 that
> $\cos (\frac{\pi}{2} \sqrt{x}) \le 1-x$, which holds for all $x \ge 0$, and consequently,
>
> $$
> | P_{K}^{\mathrm{Cos}}(\lambda) |:=
> \left|  \prod_{k=1}^K \mathrm{cos}\left( \frac{\pi}{2} \sqrt{ \frac{\lambda}{ r_{\sigma(k)}^{(K)} }  }  \right)  \right| \leq \left| \prod_{k=1}^K \left(1 - \frac{\lambda}{ r_{\sigma(k)}^{(K)} }\right)   \right| = \left| \bar{ \Phi }_K(\lambda)  \right|,
> $$
>
> The key lemma above implies that if we set the integration time as
> $ \eta_k^{(K)}
> = \frac{\pi}{2 } \frac{1}{ \sqrt{ 2 r_{\sigma(k)}^{(K)} } }$, then we get acceleration of HMC.

---

> > ### Author Response · Authors · 2022-12-08
> > **Dear Reviewer XHWf**
> >
> > Thanks for the positive feedback and comments. Please let us know if we can answer any questions. We would appreciate it. Thank you.

---

### Official Review · Reviewer_dujW · 2022-10-25

**Confidence:** 2
**Correctness:** 4
**Technical Novelty And Significance:** 2
**Empirical Novelty And Significance:** 3
**Recommendation:** 6

**Clarity, Quality, Novelty And Reproducibility:**

The paper is clearly written. The quality of the work is high. The found upper bound is novel (though, it covers a very restrictive setting) and the algorithm seems reproducible (though I have not checked the code).


**Details Of Ethics Concerns:**

No concern

**Strength And Weaknesses:**

The paper is well-written and the proofs seem correct. Though, I should highlight I did not check all the proof details closely.
The newly found bound is interesting though the studied theoretical setting, namely exact HMC (i.e. where the equations of motion are solved in closed-form) is only applicable to Gaussians and therefore, it is quite restrictive because clearly, for sampling from Gaussians there exist better algorithms and HMC is not used.
Nonetheless, this theoretical finding may open the way for future theoretical works that consider more general families of distributions.
Also, the presented empirical results show that the proposed algorithm can outperform the baseline on models other than Gaussian.


**Summary Of The Paper:**

This paper proposes a new time-varying integration duration for HMC using the roots of Chebyshev polynomials.
In a very restrictive setting, they paper proves an upper bound on the number of iterations required to reach Wasserstrain-2 distance less than an a specified error threshold. This setting is where exact HMC mechanism is applicable (i.e. the equations of motion can be solved in closed form). This closed-form solution only exists for Gaussian distributions (as far as I know).
Nonetheless, experiments shows that their scheme has benefits in case the target distribution is not Gaussian but is associated with a smooth strongly convex potential energy function.


**Summary Of The Review:**

This paper presents a new method to adapt the HMC's integration time. It is shown that on the experimental models, the proposed method can outperform the baseline.
They also find new bounds for a very restrictive case where HMC's state evolution can be solved in closed-form. As such, I am not sure if it can be called significant. Nonetheless, I think this finding is interesting and valuable.  It may also open doors to future theoretical works.

---

> ### Author Response · Authors · 2022-11-12
> **Thanks for the positive feedback and comments.**
>
> Thanks for the positive feedback and comments.
>
> Regarding going beyond quadratics, we think that there are at least three directions to try.
> - The first is to check if the analysis for the case of sampling from general strongly logconcave distributions in Chen and Vempala (2019) can be used/improved to get an accelerated rate.
> - The second is to identify when and why gradient descent with the Chebyshev step sizes of Agarwal et al. (2021) has provable acceleration beyond quadratics for optimization,
> because the intuition behind our acceleration result for HMC is based on a connection to this method. (The intuition can be found in our response to Reviewer XHWf.) Currently the acceleration result of Agarwal et al. (2021) is limited to minimizing strongly convex quadratic functions, even though they show promising empirical results compared to gradient descent with a constant step size for minimizing some non-quadratic smooth strongly convex functions. We think that identifying any conditions that gradient descent with the Chebyshev step sizes can have provable acceleration for minimizing general (or any subclasses of) smooth strongly convex functions might shed light on acceleration of HMC beyond quadratics.
> - The third is to try analyzing ideal HMC for sampling from general strongly log-concave distributions in 1-dimension. While the differential equation of ideal HMC often does not have a closed form, which we believe is why showing acceleration beyond quadratics is challenging, it is interesting to check if we can get a bound that involves the cosine product as the case of quadratic potentials (perhaps modulo some factors). If so, then we could proceed to check if acceleration is possible.
>
>
> Overall, as the reviewer kindly points out, our work may open doors to future theoretical works.
>
>
> __References:__
>
> [1] Zongchen Chen and Santosh S Vempala. Optimal convergence rate of Hamiltonian Monte Carlo for strongly logconcave distributions. International Conference on Randomization and Computation
> (RANDOM), 2019.
>
> [2] Naman Agarwal, Surbhi Goel, and Cyril Zhang. Acceleration via fractal learning rate schedules.
> ICML, 2021.

---

> > ### Author Response · Authors · 2022-12-08
> > **Dear Reviewer dujW**
> >
> > Thanks for the positive feedback and comments. Please let us know if we can answer any questions. We would appreciate it. Thank you.

---

### Official Review · Reviewer_Kxzm · 2022-10-25

**Confidence:** 4
**Correctness:** 4
**Technical Novelty And Significance:** 3
**Empirical Novelty And Significance:** 2
**Recommendation:** 5

**Clarity, Quality, Novelty And Reproducibility:**

The clarity needs to be improved, more explanations regarding why Chebyshev integration time can lead to acceleration need to be added.

Novelty and originality are acceptable, no prior works have shown similar results.


**Strength And Weaknesses:**

Strength:

* this paper proposes a novel time-varying integration method for HMC and proves a faster convergence compared to the constant-time integration method.
* this paper incorporated the proposed integration into a practical HMC algorithm and demonstrate its better performance in experiments.

weakness:
* The theory is limited to quadratic cases, while prior works cover all smooth and strong convex cases.
* The explanation of the acceleration effect of Chebyshev integration time is not clear.
* The stepsize is not tuned for HMC with constant-time integration (as mentioned in Section 4, the stepsize for HMC with constant integration time is set by (5)).

Questions:
* What’s the role of $\bar \Phi_K(\lambda)$ in the algorithm or theory? Why can equation (11) illustrate the acceleration of the proposed Chebyshev integration time?
* The authors mention that the permutation $\sigma$ is not needed in the analysis, does it mean that Theorem 1 holds for any permutation?
* (5) is only one choice of constant integration time, can we find a better choice to gain faster convergence?
* It would be better to compare the total integration time of the proposed method and baselines.
* Following the previous comment, can we fix the total integration time and directly optimize the integration time in each iteration according to Lemma 2?


**Summary Of The Paper:**

This paper proposed a time-varying integration method for accelerating the mixing of Hamiltonian Monte Carlo. In particular, when the potential $f$ is quadratic function, i.e., the target distribution is Gaussian, the ideal HMC with the proposed time-varying integration enjoy a $O(\sqrt{\kappa}\log(1/\epsilon))$ iteration complexity in terms of Wasserstein-2 distance. The authors further develop a practical HMC algorithm using the proposed integration time. Experimental results show that the developed algorithm can also lead to better sampling performance for the general smooth and strongly convex potentials.

**Summary Of The Review:**

My major concern is (1) the limitation of theory (i.e., only for quadratic potential) and (2) lack of thorough discussion and exploration of baseline method (e.g., constant integration time) in terms of both theory and experiment.

---

> ### Author Response · Authors · 2022-11-12
> **Thanks for the comments and feedback (1/3)**
>
> Thanks for the comments and feedback. Please kindly find our response to each of the comments below.
>
> __*1. The theory is limited to quadratic cases, while prior works cover all smooth and strong convex cases.*__
>
> We would like to clarify that prior theory works of Hamiltonian Monte Carlo for general strongly log-concave distributions only achieved a non-accelerated rate $1-\Theta\left( \frac{1}{\kappa} \right)  $, while our work is about accelerating Hamiltonian Monte Carlo and we prove a solid accelerated rate $1-\Theta\left( \frac{1}{ \sqrt{\kappa} } \right)  $ for the strongly convex quadratic case.
>
>
> Hamiltonian Monte Carlo is a classical sampling method proposed more than three decades ago,
> and how to accelerate the method with strong guarantees has been a big open question in the literature.
> We are very excited that the current work has made a solid step forward on accelerating this classical method. While our theoretical result is currently limited to the strongly convex quadratic potential, the experiments of sampling from general strongly log-concave distributions show that the proposed scheme of choosing the integration time accelerates HMC compared to the constant integration time. Hence, our work might help facilitate future research in accelerating HMC for sampling from general strongly log-concave distributions with strong theoretical guarantees, and we would like to pursue this direction in the future.
>
> __*2. The explanation of the acceleration effect of Chebyshev integration time is not clear.*__
>
>
> Thanks for the comment. While we briefly mentioned Chebyshev iterations and gradient descent with the Chebyshev step sizes (Agarwal et al. (2021)) and stated that our acceleration result can be viewed as a counterpart from optimization in the paper, we would like to provide an intuition behind our acceleration result here.
> But before that, we would like to describe the work of gradient descent with the Chebyshev step sizes (Agarwal et al. (2021)) in more detail, because we are going to draw a connection between optimization and sampling to showcase the intuition.
> Agarwal et al. (2021) provably show that gradient descent with a scheme of step sizes based on the Chebyshev Polynomials has an accelerated rate for minimizing strongly convex quadratic functions compared to GD with a constant step size, and their experiments show some promising results for minimizing smooth strongly convex functions beyond quadratics via the proposed scheme of step sizes.
> More precisely, define $f(w)= \frac{1}{2} w^{\top} A w$, where $A \in \mathbb{R}^{{d \times d}}$ is a positive definite matrix which has eigenvalues
> $L:=\lambda_{1} \geq \lambda_{2} \geq \dots \geq \lambda_{{d}}=:m$.
> Agarwal et al.\ (2021) consider applying gradient descent
> $$ w_{k+1} = w_k - \eta_k \nabla f(w_k)$$
> to minimize $f(\cdot)$,
> where $\eta_{k}$ is the step size of gradient descent at iteration $k$.
> Let $w_{\*}$ be the unique global minimizer of $f(\cdot)$.
> It is easy to show that the dynamic of the distance evolves as
> $$
> w_{k+1}-w_* = (I_d - \eta_k A) (I_d - \eta_{k-1} A) \cdots (I_d - \eta_{1} A) (w_1 - w_*).$$
> Hence, the size of the distance to $w_*$ at iteration $K+1$ is bounded by
>
> $$\\\| w_{K+1} - w_* \\\| \leq \max_{ j \in [d] } | \prod_{k=1}^K  ( 1 - \eta_k \lambda_j ) | \\\| w_1  - w_* \\\|.$$
>
> This shows that the convergence rate of GD is governed by
> $\max_{ j \in [d] } | \prod_{k=1}^K  ( 1 - \eta_k \lambda_j ) | $.
> By setting $\eta_{k}$ as the inverse of the Chebyshev root
> $r_{k}^{(K)}$ or any permuted root $r_{ \sigma(k) }^{(K)}$ (see eq. (8) in our paper for the definition),
> the polynomial
> $ \prod_{k=1}^K  ( 1 - \eta_k \lambda ) $ is actually
> the $K$-degree scale-and-shifted polynomial, i.e.,
> $ \prod_{k=1}^K  ( 1 - \eta_k \lambda ) = \prod_{k=1}^K \left( 1-\frac{\lambda}{ r_{\sigma(k)}^{(K)} } \right) = \bar{{\Phi}}_{k}(\lambda) $
>
> (see eq. (7) in our paper for the definition).
> It is well-known in the literature of optimization and numerical linear algebra that
> the $K$-degree scale-and-shifted polynomial satisfies
>
> $$ \max_{\lambda \in [m,L]} \left| \bar{\Phi}_K(\lambda) \right| \leq 2 \left( 1 - 2 \frac{ \sqrt{m} }{ \sqrt{L} + \sqrt{m} } \right)^{K} = O\left( \left( 1 - \Theta\left( \sqrt{ \frac{m}{L} } \right)\right)^K  \right),$$
>
> which is restated in Lemma 3 of our paper and its proof is replicated in Appendix A of our paper for the reader's convenience. Applying this result, one gets a simple proof of the accelerated linear rate of GD with the proposed scheme of step sizes for minimizing quadratic functions. A nice blog article by Pedregosa (2021) explains this in detail.
>
> Now we are ready to highlight its connection with HMC.
> (to be described in the next block)

---

> > ### Author Response · Authors · 2022-11-12
> > **Thanks for the comments and feedback (2/3)**
> >
> > (Continue)
> >
> > Now we are ready to highlight its connection with HMC.
> > In Lemma 2 of the paper, we restate a known result in HMC literature,
> > where its proof is also replicated in Appendix A of the paper for the reader's convenience.
> > The lemma states that:
> >
> > *For sampling distributions with strongly convex quadratic potentials, the 2-Wasserstein distance
> > to the target distribution $\pi$ evolves as
> > $$W_{2}(\rho_K, \pi) \leq
> > \max_{j \in [d]} \left| \prod_{k=1}^K \mathrm{cos}\left( \sqrt{2 \lambda_j} \eta_k^{(K)} \right) \right|  W_{2}( \rho_0,\pi),$$
> > where $\rho_{K}$ is the distribution of samples generated from ideal HMC at iteration $K$ and $\\{ \lambda_{j} \\}_{{j=1}}^{d}$ are the eigenvalues of the quadratic matrix.*
> >
> > It is noted that the lemma holds for any scheme of integration time $\\{ \eta_k^{(K)} \\}_{k=1}^K$.
> >
> > The lemma indicates that the convergence rate of HMC is governed by $\max_{j \in [d]} | \prod_{k=1}^K \mathrm{cos}( \sqrt{2 \lambda_j} \eta_k^{(K)} ) |$. By way of comparison to that
> > of GD for minimizing quadratic functions,
> > i.e., $\max_{ j \in [d] }
> > | \prod_{k=1}^K  ( 1 - \eta_k \lambda_j ) | $, it appears that they share some similarity.
> > Specifically, if we could upper-bound each cosine term in the product as
> > $\mathrm{cos}\left( x \right) \leq 1 -x$ and replace the cosine term with the upper bound, then the product of cosine terms that governs the progress of HMC becomes
> > the $K$-degree scale-and-shifted Chebyshev Polynomial when we choose the integration time $\eta_k^{(K)}$ appropriately,
> > which means that we get the accelerated linear rate of HMC as the consequence.
> > A caveat of this intuition is that
> > the inequality $\mathrm{cos}\left( x \right) \leq 1 -x$ only holds for small $x$.
> > That is, it breaks when $x$ is large, and so we cannot use this inequality to prove acceleration.
> > Fortunately, we are able to show another inequality
> > $\cos (\frac{\pi}{2} \sqrt{x}) \le 1-x$ which holds for all $x \ge 0$, and consequently,
> > $$
> > | P_{K}^{\mathrm{Cos}}(\lambda) |:=
> > \left|  \prod_{k=1}^K \mathrm{cos}\left( \frac{\pi}{2} \sqrt{ \frac{\lambda}{ r_{\sigma(k)}^{(K)} }  }  \right)  \right| \leq \left| \prod_{k=1}^K \left(1 - \frac{\lambda}{ r_{\sigma(k)}^{(K)} }\right)   \right| = \left| \bar{ \Phi }_K(\lambda)  \right|,
> > $$
> >
> > which is what equation (11)/Lemma 4 states in the paper. The proof of Lemma 4 is in Appendix A of the paper.
> > The key lemma above implies that if we set the integration time as
> > $ \eta_k^{(K)}
> > = \frac{\pi}{2 } \frac{1}{ \sqrt{ 2 r_{\sigma(k)}^{(K)} } }$, then we get acceleration of HMC.
> >
> > We hope that we have now clearly described the intuition behind our acceleration result. Please kindly let us know if there exists any missing part that we have to elaborate.
> >
> >
> > __References:__
> >
> > [1]
> > Naman Agarwal, Surbhi Goel, and Cyril Zhang. Acceleration via fractal learning rate schedules.
> > ICML, 2021.
> >
> > [2]
> > Fabian Pedregosa. Acceleration without momentum, 2021.
> >
> > __*3. The step size is not tuned for HMC with constant-time integration (as mentioned in Section 4, the step size for HMC with constant integration time is set by (5)).*__
> >
> > We would like to clarify that the constant integration time has been tuned.
> > Moreover, in Section 2.2, we state a known lower bound of using any constant integration time in the literature, which is $\Omega(\kappa)$ (Chen and Vempala (2019)). For the upper-bound result, the tutorial by Vishnoi (2021) provides a simple proof, from which one can see that the constant integration time (5) is the best.
> >
> > __References:__
> >
> > [1] Zongchen Chen and Santosh S Vempala. Optimal convergence rate of Hamiltonian Monte Carlo for strongly logconcave distributions. International Conference on Randomization and Computation (RANDOM), 2019.
> >
> > [2] Nisheeth K. Vishnoi.
> > An introduction to Hamiltonian Monte Carlo method for sampling.
> > arXiv:2108.12107, 2021.
> >
> >
> > __*4. What's the role of $\bar{\Phi}_{K}(\lambda)$ in the algorithm or theory? Why can equation (11) illustrate the acceleration of the proposed Chebyshev integration time?*__
> >
> > Thanks for raising this question. Please kindly find our response to the comment
> > ``The explanation of the acceleration effect of Chebyshev integration time is not clear.'' above, where we have answered this question in detail.
> >
> > __*5. The authors mention that the permutation is not needed in the analysis, does it mean that Theorem 1 holds for any permutation?*__
> >
> > Yes, Theorem 1 holds for any permutation of the use of the Chebyshev roots, which can also be seen from its proof on Page 6 in the paper.
> >
> > To see this, observe that the value of the $K$-degree polynomial
> > $\prod_{k=1}^K \left(1 - \frac{\lambda}{ r_{\sigma(k)}^{(K)} }\right)$
> >  is always the same for any permutation $\sigma(\cdot)$.

---

> > > ### Author Response · Authors · 2022-11-12
> > > **Thanks for the comments and feedback (3/3)**
> > >
> > > __*6. (5) is only one choice of constant integration time, can we find a better choice to gain faster convergence?*__
> > >
> > > Thanks for raising the question. The answer in no.
> > > In Section 2.2, we state a known lower bound of using any constant integration time in the literature, which is $\Omega(\kappa)$ (Chen and Vempala (2019)). For the upper-bound result, Vishnoi (2021) has provided a simple proof in an excellent tutorial, one can see from his note that the constant integration time (5) is the best.
> > >
> > > __*7 It would be better to compare the total integration time of the proposed method and baselines. Following the previous comment, can we fix the total integration time and directly optimize the integration time in each iteration according to Lemma 2?*__
> > >
> > > Thanks for raising this question.
> > > As the reviewer points out, the integration time in each iteration $k$ is different.
> > > While the squared integration time in iteration $k$
> > > using the proposed scheme of integration time
> > > is $\frac{\pi^2}{8} \frac{1}{r_{\sigma(k)}^{(K)}}$,
> > > the squared integration time in iteration $k$ of the $\frac{\pi^2}{8} \frac{1}{L}$
> > > of the constant integration time.
> > > This means that the average of squared integration time is
> > > $  \frac{1}{K}  \sum_{k=1}^K \frac{\pi^2}{8} \frac{1}{r_{\sigma(k)}^{(K)}}
> > > $ of the proposed scheme of the integration time, while it is $\frac{\pi^2}{8} \frac{1}{L}$
> > > of the constant integration time.
> > > We consider squared integration time, because
> > > Item 3 in Proposition~6 of Agarwal et al.\ (2021) has established that
> > > $$
> > > \frac{1}{K} \sum_{k=1}^K \frac{1}{r_{\sigma(k)}^{(K)}}
> > > = \frac{ \tanh( K \mathrm{arccosh}( \frac{2m}{L-m} ) ) }{ \sqrt{Lm} }
> > > < \frac{\sqrt{\kappa}}{L}. %\quad \clubsuit
> > > $$
> > >
> > > Now recall that to get an $\epsilon$ error in 2-Wasserstein distance, the total number of ideal HMC steps is
> > > $\sqrt{\kappa} \log \left( \frac{1}{\epsilon}  \right) $ using the proposed scheme
> > > for the integration time, while the total number of ideal HMC steps is
> > > $\kappa \log \left( \frac{1}{\epsilon}  \right) $ using the best constant integration time. If we define
> > >  the computational consumption to get an $\epsilon$ error
> > > as the number of HMC steps times the average of squared integration time,
> > > then we have
> > > $$ \text{ computational consumption by HMC with Chebyshev integration time }
> > > = \sqrt{\kappa} \log \left( \frac{1}{\epsilon}  \right)
> > > \times
> > > \frac{\pi^2}{8} \frac{1}{K}  \sum_{k=1}^K \frac{1}{r_{\sigma(k)}^{(K)}},
> > > $$
> > > and
> > > $$
> > > \text{ computational consumption by HMC with constant integration time }
> > > = \kappa \log \left( \frac{1}{\epsilon}  \right) \times \frac{\pi^2}{8} \frac{1}{L}.
> > > $$
> > > Combining the above and Item 3 in Proposition~6 of Agarwal et al.\ (2021), we see that
> > >  using the Chebyshev integration time allows HMC to
> > > get an $\epsilon$ error with less computational consumption. In other words, if we fix the total (squared) integration time, then HMC with the Chebyshev integration time would allow to get a smaller 2-Wasserstein distance than that of using the best constant integration time.
> > >
> > > __Reference__
> > >
> > > Naman Agarwal, Surbhi Goel, and Cyril Zhang. Acceleration via fractal learning rate schedules.
> > > ICML, 2021.
> > >
> > > __*8. My major concern is (1) the limitation of theory (i.e., only for quadratic potential) and (2) lack of thorough discussion and exploration of baseline method (e.g., constant integration time) in terms of both theory and experiment.*__
> > >
> > > Please kindly find our response above to check if we have clarified all the concerns.
> > > As we mentioned, Hamiltonian Monte Carlo is a classical sampling method,
> > > and how to accelerate the method with strong theoretical guarantees has been a challenging open question for years. We are excited about making a solid step in this direction. Though our acceleration result is currently proven only in the case of quadratic potentials, experiments of sampling from general strongly log-concave distributions have shown its promising power in practice, which opens doors to investigate when and why HMC with the Chebyshev integration time can have a provable acceleration for sampling from general strongly log-concave distributions.
> > >
> > > We have also showcased the intuition of our acceleration result in this rebuttal.
> > > Given that ours is the first to show that the rate $1- \Theta\left( \frac{1}{\sqrt{\kappa}} \right)$ of HMC can be achieved and that a (non-acceleration) lower bound of HMC via any constant integration time
> > > has been established in the literature, we believe that the comparison with the baseline (HMC with the constant integration time) is adequate on both the theoretical side and the experimental side.
> > > In the experiments, we have compared the proposed Chebyshev integration time with the baseline in terms of 4 metrics on 5 sampling tasks.
> > >
> > > We will appreciate it if the reviewer can consider upgrading the score (if we have written a satisfying rebuttal). We will also appreciate it if the reviewer can let us know if there exists any lingering concern, and we would be eager to clarify it. Thank you.

---

> ### Author Response · Authors · 2022-11-18
> **Comparison of the total integration time (1/2)**
>
> __*It would be better to compare the total integration time of the proposed method and baselines. Following the previous comment, can we fix the total integration time and directly optimize the integration time in each iteration according to Lemma 2?*__
>
> Previously we answered the reviewer's question by comparing the total squared integration time, because there seems no simple closed-form of the total (non-squared) integration time for HMC with the Chebyshev scheme.
> However, while the exact form is hard to obtain, a good approximation of the total integration time can be established, which is described in detail in the following.
> Consequently, we can compare the total integration time as the reviewer's request.
>
> Recall the number of iterations to get an $\epsilon$ 2-Wasserstein distance to the  target distribution is $K=O\left( \sqrt{\kappa} \log\left( \frac{1}{\epsilon} \right) \right)$ of HMC with the Chebyshev integration time (Theorem 1 in the paper).
> The average of the integration time is
>
> $$
> \frac{1}{K} \sum_{k=1}^K \eta_k^{(K)} =
> \frac{1}{K} \sum_{k=1}^K  \frac{\pi}{2 \sqrt{2} } \frac{1}{ \sqrt{ r_{\sigma(k)}^{(K)}} } =
> \frac{1}{K} \sum_{k=1}^K  \frac{\pi}{2 \sqrt{2} } \frac{1}{ \sqrt{ r_{k}^{(K)}} },
> $$
>
> where we recall
> $$
> \textbf{ (Chebyshev roots) } \qquad
> r_{k}^{(K)} := \frac{L+m}{2} - \frac{L-m}{2} \text{cos}\left( \frac{(k-\frac{1}{2}) \pi}{K}  \right),
> $$
> see e.g., (8) in the paper and observe that a permutation $\sigma(\cdot)$ does not affect the average.
>
> Then, if $K$ is even, we can rewrite the averaged integration time as
> $$
> \frac{1}{K} \sum_{k=1}^K \eta_k^{(K)} =
> \frac{1}{K} \frac{\pi}{2 \sqrt{2} }  \sum_{k=1}^{K/2} \left(
> \frac{1}{ \sqrt{ r_{k}^{(K)}} } +  \frac{1}{ \sqrt{ r_{K+1-k}^{(K)}} }
>     \right).
> $$
> Otherwise, $K$ is odd, we can rewrite the averaged integration time as
> $$
> \frac{1}{K} \sum_{k=1}^K \eta_k^{(K)} =
> \frac{1}{K} \frac{\pi}{2 \sqrt{2} }
> \left(
> \frac{1}{ \sqrt{ r_{(K+1)/2}^{(K)}} }
> +
>  \sum_{k=1}^{(K-1)/2} \left(
> \frac{1}{ \sqrt{ r_{k}^{(K)}} } +  \frac{1}{ \sqrt{ r_{K+1-k}^{(K)}} }
>     \right)
> \right).
> $$
>
> We will show
> $$
> \heartsuit:
> \frac{1}{ \sqrt{ r_{k}^{(K)}} } +  \frac{1}{ \sqrt{ r_{K+1-k}^{(K)}} }
> \leq \frac{1}{ \sqrt{ r_{ \lfloor K/2 \rfloor }^{(K)}} } +  \frac{1}{ \sqrt{ r_{K- \lfloor K/2 \rfloor + 1  }^{(K)}} },
> $$
> for any $k = \{1,2, \dots, \lfloor \frac{K}{2} \rfloor \}$ soon.
> Given this, we can further upper-bound the averaged integration time as
> $$
> \frac{1}{K} \sum_{k=1}^K \eta_k^{(K)} \leq
> \frac{\pi}{4 \sqrt{2} }
> \left(  \frac{1}{ \sqrt{ r_{ \lfloor K/2 \rfloor }^{(K)}} } + \frac{1}{ \sqrt{ r_{K - \lfloor K/2 \rfloor+1}^{(K)}} } \right),
> $$
> when $K$ is even, and when $K$ is odd, we can upper-bound
> the averaged integration time as
> $$
> \frac{1}{K} \sum_{k=1}^K \eta_k^{(K)} \leq
> \frac{1}{K}
> \frac{\pi}{2 \sqrt{2} }
> \left(
> \frac{1}{ \sqrt{ r_{(K+1)/2}^{(K)}} }
> +
> \frac{K-1}{2}
> \left(  \frac{1}{ \sqrt{ r_{\lfloor K/2 \rfloor}^{(K)}} } + \frac{1}{ \sqrt{ r_{K-\lfloor K/2 \rfloor+1}^{(K)}} } \right) \right).
> $$
> Using the definition of the Chebyshev root, we have
> $$
> r_{\lfloor K/2 \rfloor }^{(K)} =
> \frac{L+m}{2} - \frac{L-m}{2} \text{cos}\left( \frac{ \left( \lfloor \frac{K}{2} \rfloor - \frac{1}{2} \right) \pi}{K}  \right)
> \approx \frac{L+m}{2},
> $$
> where the approximation is because
> $\frac{ \left( \lfloor \frac{K}{2} \rfloor - \frac{1}{2} \right)   \pi}{K} \approx \frac{\pi}{2}$ when $K$ is large,
> and hence
> $\text{cos}\left( \frac{ \left( \lfloor \frac{K}{2} \rfloor - \frac{1}{2} \right)  \pi}{K}  \right) \approx 0$.
> Similarly, we can approximate
> $$
> r_{K - \lfloor K/2 \rfloor + 1 }^{(K)} =
> \frac{L+m}{2} - \frac{L-m}{2} \text{cos}\left( \frac{  \left( K - \lfloor K/2 \rfloor + 1 - \frac{1}{2} \right) \pi}{K}  \right)
> \approx \frac{L+m}{2}
> $$
> as
> $\frac{ \left( K - \lfloor K/2 \rfloor + 1 - \frac{1}{2} \right)  \pi}{K} \approx \frac{\pi}{2}$ when $K$ is large.
> Also, we can approximate
> $r_{(K+1)/2}^{(K)} \approx \frac{L+m}{2}$ when $K$ is odd and large for the same reason.
>
> Combining the above, the total integration time of HMC with the Chebyshev scheme can be approximated as
>
> $$\text{number of iterations} \times \text{average integration time} = \sqrt{\kappa} \log\left( \frac{1}{\epsilon} \right) \times \frac{1}{K} \sum_{k=1}^K \eta_k^{(K)} \approx \sqrt{\kappa} \log\left( \frac{1}{\epsilon} \right) \times \frac{\pi}{2 } \frac{ 1 }{ \sqrt{L+m} }.$$
>
>
> When $\kappa:=\frac{L}{m}$ is large,
> the total integration time becomes
> $$
> \clubsuit:
> \sqrt{\kappa} \log\left( \frac{1}{\epsilon} \right)
> \times
> \frac{\pi}{2 }
> \frac{ 1 }{ \sqrt{L+m} }
> = \Theta \left( \frac{1}{\sqrt{m}} \log\left( \frac{1}{\epsilon} \right) \right).
> $$
>
> (to continue)

---

> > ### Author Response · Authors · 2022-11-18
> > **Comparison of the total integration time (2/2)**
> >
> > (Continue)
> >
> > Now let us switch to analyzing HMC with the best constant integration time $\eta = \Theta\left( \frac{1}{\sqrt{L}} \right) $ ((5), Vishnoi (2021)), which has
> > the non-accelerated rate.
> > Specifically, it needs
> > $K=O\left( \kappa \log\left( \frac{1}{\epsilon} \right) \right)$
> > iterations to converge to the target distribution.
> > Hence,
> > the total integration time of HMC with the best constant integration time is
> > $$\spadesuit:
> > \text{ number of iterations} \times \text{average integration time} = \kappa \log\left( \frac{1}{\epsilon} \right) \times \Theta \left( \frac{ 1 }{ \sqrt{L} } \right) = \Theta \left( \frac{\sqrt{L}}{m} \log\left( \frac{1}{\epsilon} \right) \right).$$
> > By way of comparison ($\clubsuit$ vs.~$\spadesuit$),  __we see that the total integration time of HMC with the proposed scheme of Chebyshev integration time reduces by a factor $\sqrt{\kappa}$,
> > compared with HMC with the best constant integration time.__
> >
> >
> > The remaining thing to show is the inequality
> >
> > $$\heartsuit: \frac{1}{ \sqrt{ r_{k}^{(K)}} } +  \frac{1}{ \sqrt{ r_{K+1-k}^{(K)}} }
> > \leq \frac{1}{ \sqrt{ r_{ \lfloor K/2 \rfloor }^{(K)}} } +  \frac{1}{ \sqrt{ r_{K+1- \lfloor K/2 \rfloor  }^{(K)}} },$$
> >
> > for any $k = \{1,2, \dots, \lfloor \frac{K}{2} \rfloor \}$.
> >
> > We have
> > $$ \frac{1}{ \sqrt{ r_{k}^{(K)}} } +  \frac{1}{ \sqrt{ r_{K+1-k}^{(K)}} } = \sqrt{2} \times \left(  \frac{1}{  \sqrt{ L+m - (L-m) \mathrm{cos}\left( \frac{ \left( k - \frac{1}{2} \right) \pi}{K}  \right)    }}+\frac{1}{ \sqrt{L+m - (L-m) \mathrm{cos}\left( \frac{  \left( K-k+\frac{1}{2} \right) \pi}{K}  \right)    }}   \right)
> >  = \sqrt{2} \times \left(
> > \frac{1}{ \sqrt{ L+m - (L-m) \mathrm{cos}\left( \frac{ \left( k - \frac{1}{2} \right) \pi}{K}  \right)    }}+\frac{1}{ \sqrt{ L+m + (L-m) \mathrm{cos}\left( \frac{  \left( k - \frac{1}{2} \right) \pi}{K}  \right)   }}   \right)$$
> >
> >
> >
> > Now let us
> > define
> > $H(k):=
> > \left(
> > \frac{1}{
> > \sqrt{
> > L+m - (L-m) \mathrm{cos}\left( \frac{ \left( k - \frac{1}{2} \right) \pi}{K}  \right)
> > }
> > }
> > +
> > \frac{1}{
> > \sqrt{
> > L+m + (L-m) \mathrm{cos}\left( \frac{ \left( k - \frac{1}{2} \right) \pi}{K}  \right)
> > }
> > }
> >    \right)$ and treat $k$ as a continuous variable.
> >
> > The derivative of $H(k)$ is
> >
> > $$H'(k)  = \frac{\pi}{2 K} (L-m) \mathrm{sin}\left( \frac{ \left( k-\frac{1}{2} \right) \pi}{K}  \right) \times \left( \frac{1}{  \left(   L+m - (L-m) \mathrm{cos}\left( \frac{ \left( k-\frac{1}{2} \right) \pi}{K}  \right)           \right)^{3/2} } - \frac{1}{ \left(   L+m + (L-m) \mathrm{cos}\left( \frac{\left( k-\frac{1}{2} \right) \pi}{K}  \right)    \right)^{3/2} } \right) > 0.$$
> >
> > That is, $H'(k)$ is an increasing function of $k$ when  $1\leq k \leq \lfloor \frac{K}{2} \rfloor$, which implies
> >  that the inequality $\heartsuit$. Now we have completed the analysis.
> >
> > ===
> >
> > The reviewer's suggestion helps improve the presentation/interpretation of our result. We highly appreciate it.
> >
> > We have addressed each of the comments of the reviewer. It would be appreciated if the reviewer can let us know if we have provided satisfying answers and responses, and consider upgrading the score. Or, it would also be appreciated if the reviewer can let us know if we can address any parts in more detail. Thank you.

---

> ### Author Response · Authors · 2022-11-28
> **Dear Reviewer Kxzm**
>
> Dear Reviewer Kxzm,
>
> We would very much appreciate it if the reviewer can engage with us, and kindly let us know if we have clarified all the concerns and provided satisfying answers. Or, let us know if we should address any parts in more detail, and we will be happy to do so.  Thank you.
>
> All the best,
>
> The authors

---

> > ### Author Response · Authors · 2022-12-03
> > **Dear Reviewer Kxzm**
> >
> > Sorry for sending this message again. But we would very much appreciate it if the reviewer can engage with us, and kindly let us know if we have clarified all the concerns and provided satisfying answers. Or, let us know if we should address any parts in more detail, and we will be happy to do so. Thank you.
> >
> > All the best,
> >
> > The authors

---

### Official Review · Reviewer_GRbp · 2022-10-25

**Confidence:** 3
**Correctness:** 4
**Technical Novelty And Significance:** 3
**Empirical Novelty And Significance:** 2
**Recommendation:** 5

**Clarity, Quality, Novelty And Reproducibility:**

The paper is quite clear. I did not ceck the experiments carefully, but the code is well-documented and "looks reproducible". The approach also seems novel in this context.

I do not think the paper is of great interest, given that the theory is severely limited, and the experiments do not provide a clear indication of practical aspects of this method. One issue I would like to see explored is this: if the likelihood is complicated, maybe it's hard to compute $\kappa$. What does one do in practice? Is the method sensitive to the upper bound $L$ and the lower bound $m$?

**Strength And Weaknesses:**

*Strengths*

- This seems to be the first acceleration result for HMC.
- The use of Chebyshev polynomials is somewhat interesting.

*Weaknesses*

- Unfortunately, the theory does not cover any cases of practical interest.
- The experimental success metrics are all defined in terms of effective sample size (ESS). This is a useful proxy of MCMC quality, but it does not prove convergence, nor does it quantify this convergence in harder problems. It would be nice to see other metrics as in: can one approximate the integrals of some "interesting/hard functions" from the MC trajectories of different methods?
- In line with the previous remark, it'd be nice to look at some *hard* distributions coming from real-life Bayesian statistics. What do the credible regions look like? Does one reobtain known results at lower cost?



**Summary Of The Paper:**

The paper considers the possibility of accelerating Hamiltonian Monte Carlo (HMC) methods for sampling from distributions $\pi$. For a $L$-smooth and $m$-convex $f$, the complexity of sampling from $\pi \;\alpha\; e^{-f}$ (via an "ideal method" that does not discretize time) is of the order $\kappa \log(1/\delta)$, where $\delta$ is the desired precision and $\kappa={L/m}$ is the condition number. The present paper shows that one can lower the complexity of the ideal method by a factor of $\sqrt{\kappa}$ when $f$ is quadratic (and so $\pi$ is Gaussian). This requires knowledge of $L$ and $m$, which one uses to choose special step sizes defined via Chebyshev polynomials. Experiments suggest that the same choice of weights works beyond the case of quadratic $f$; in fact the method seems to "beat the competition" even for nonconvex likelihoods.

**Summary Of The Review:**

The paper has a nice idea, but, at it is present form, its interest is too limited from both theoretical and practical perspectives.

UPDATE on Nov 30th

I have increased my "novelty" and overall scores. I thank the authors for their responses. To comment on a few points.

a) I agree ESS is a standard surrogate for convergence. It would be nice, however, to see a problem with eg. a multimodal likelihood and show (empirically) that accelerated HMC converges faster than the standard method.

b) I understand that the above point is (as the authors say) disjoint from their goals, as it'd go beyong the logconcave setting. However, given the lack of theoretical results for general logconcave f, I think the paper would greatly benefit from a more extensive empirical evaluation of the accelerated method in more practical settings.

c) I thank the authors for the clarifiction regarding sensibility to the smoothness ans strong convexity parameters.

---

> ### Author Response · Authors · 2022-11-12
> **Thanks for the comments. (1/3)**
>
> Thanks for the comments and feedback. Please kindly find our response to each of the comments below.
>
> 1. __*"Unfortunately, the theory does not cover any cases of practical interest". "I do not think the paper is of great interest, given that the theory is severely limited"*__
>
> The reviewer is correct that our theory currently only covers the quadratic case. However, we point out that the quadratic case is an important setting that provides evidence whether we can apply the proposed technique with provable results.
> For example, in the paper we mentioned Chebyshev iteration and the work of Gradient descent with the Chebyshev step size (Agarwal et al. 2021), which also only covers the quadratic case.
> Similarly, the classical paper of the Heavy Ball momentum method (Polyak 1964) ``only'' showed acceleration for minimizing strongly convex quadratic functions, but it became a foundation of further development both for theoretical results and practical algorithms.
>
> Hamiltonian Monte Carlo (HMC) is a classical sampling method proposed more than three decades ago, which is very popular in practice. The question of how to accelerate sampling algorithms in parallel to optimization is an important open question that has received much recent attention, but not a concrete resolution yet. We are very excited that our current work has made a solid step forward on accelerating this classical method. While our theoretical result is currently limited to the strongly convex quadratic potential, the experiments of sampling from general strongly log-concave distributions show that the proposed scheme of choosing the integration time accelerates HMC compared to the constant integration time. Hence, our work might help facilitate future research in accelerating HMC for sampling from general strongly log-concave distributions with strong theoretical guarantees, and we would like to pursue this direction in the future.
>
> __References:__
>
> [1] Naman Agarwal, Surbhi Goel, and Cyril Zhang. Acceleration via fractal learning rate schedules. ICML, 2021.
>
> [2] B.T. Polyak. Some methods of speeding up the convergence of iteration methods. USSR Computational Mathematics and Mathematical Physics, 1964.
>
>
>
> 2. __*... the experiments do not provide a clear indication of practical aspects of this method.*__
>
> We would like to clarify that on Page 6 of this paper, we provide a clear description on how to use HMC with the proposed scheme of the Chebyshev integration time to sample from general strongly log-concave distributions; see Algorithm 2 on Page 6. Furthermore, in the experiments, we consider sampling from three non-quadratic strongly logconcave distributions (mixture of Gaussians, Bayesian logistic regression, and a hard distribution proposed in Lee et al. (2021)). The experimental results show that HMC with the proposed integration time outperforms HMC with the constant integration time even for the task of sampling from general strongly log-concave distributions. Therefore, we believe that our paper did provide a clear indication of practical aspects of the proposed method.

---

> ### Author Response · Authors · 2022-11-12
> **Thanks for the comments. (2/3)**
>
> 3. ___*The experimental success metrics are all defined in terms of effective sample size (ESS). This is a useful proxy of MCMC quality, but it does not prove convergence, nor does it quantify this convergence in harder problems. It would be nice to see other metrics as in: can one approximate the integrals of some "interesting/hard functions" from the MC trajectories of different methods?*___
>
>
> Effective sample size is one of the most popular metric for evaluating and comparing different sampling methods in sampling literature, see e.g, (the references we listed below) Girolami et al. (2011); Brofos \& Lederman (2021); Hirt et al. (2021); Riou-Durand \& Vogrinc (2022);
> Hoffman et al.\ (2021); Hoffman \& Gelman (2014); Steeg \& Galstyan (2021); Kook et al. (2022). We also note that whether there exists a better surrogate than the effective sample size for evaluating the qualities of samples
> and the convergence speed of an underlying sampling method is an open question in sampling literature, given that distance metrics like Wasserstein distance are often difficult to be reliably estimated in practice.
>
> In our experiments of sampling from a Gaussian distribution, we have considered
> two additional metrics other than ESS. Specifically,
> in Figure 2 in the paper we plot the difference between the estimated covariance and the true covariance as well as a surrogate metric which we call discrete TV over iterations. We found that HMC with the proposed scheme of the integration time also significantly outperforms the baseline under these two metrics.
>
> Second, to the best of our knowledge, the metric of
> ``approximating the integrals of some interesting/hard functions''
> proposed by the reviewer has not been adopted in the literature for comparing convergence rates of different sampling algorithms, and it is not clear how the proposed metric is meaningful for solving the issue of ESS or comparing the convergence to a target distribution of HMC.
>
>
> ___References:___
>
> [1] Mark Girolami, Ben Calderhead, Siu A. Chin. Riemann manifold Langevin and Hamiltonian Monte Carlo methods. Journal of the Royal Statistical Society, 2011.
>
> [2] James A. Brofos and Roy R. Lederman. Evaluating the implicit midpoint integrator for Riemannian manifold Hamiltonian Monte Carlo. ICML, 2021.
>
> [3] Marcel Hirt, Michalis K. Titsias, and Petros Dellaportas. Entropy-based adaptive Hamiltonian Monte Carlo. NeurIPS, 2021.
>
> [4] Lionel Riou-Durand and Jure Vogrinc. Metropolis Adjusted Langevin trajectories: a robust alternative to Hamiltonian Monte Carlo. arXiv:2202.13230, 2022.
>
> [5] Matthew D. Hoffman, Alexey Radul, and Pavel Sountsov. An adaptive-MCMC scheme for setting trajectory lengths in Hamiltonian Monte Carlo. AISTATS, 2021.
>
> [6] Greg Ver Steeg and Aram Galstyan. Hamiltonian dynamics with non-newtonian momentum for rapid sampling. NeurIPS, 2021.
>
> [7] Yunbum Kook, Yin Tat Lee, Ruoqi Shen, Santosh S. Vempala. Sampling with Riemannian Hamiltonian Monte Carlo in a Constrained Space. NeurIPS, 2022.
>
> ___*4. In line with the previous remark, it'd be nice to look at some hard distributions coming from real-life Bayesian statistics. What do the credible regions look like? Does one reobtain known results at lower cost?*___
>
> We believe these questions are disjoint from this work. Our work is about sampling, more precisely, about accelerating HMC which is a classical sampling method. Our work does not concern the aspects of statistical modeling or estimation, or credible regions.
>
> ___*5. One issue I would like to see explored is this: if the likelihood is complicated, maybe it's hard to compute . What does one do in practice?*___
>
> Similar to the above, we believe this question is disjoint from our work. We address the problem of sampling when we are given access to the gradient of the log-density. Computing likelihood, e.g. in Bayesian model, is a different problem that we do not study in this paper.

---

> ### Author Response · Authors · 2022-11-12
> **Thanks for the comments (3/3)**
>
> __*6. Is the method sensitive to the upper bound L and the lower bound m?*__
>
> Thanks for raising this question.
> On the theoretical side,
> recall that our acceleration result relies on
> the scale-and-shifted Chebyshev polynomial,
> there are works in the literature studying about how noise-tolerant of the scale-and-shifted Chebyshev polynomial is for getting an accelerated rate
> when $L$ and $m$ are not precise, see e.g., Agarwal et al. 2021 and the references therein. On the empirical side, our experiments of sampling from general strongly log-concave distributions shows that the acceleration of HMC via the proposed scheme
> is not sensitive to the estimate of $L$ and $m$. Indeed, as described in the setup, for sampling from these general strongly log-concave distributions, we simply use the largest eigenvalue and the smallest eigenvalue of the Hessian at the minimizer of the potential as an estimate/surrogate of the smoothness constant $L$ and the strong convexity constant $m$ respectively.
>
> __*7.   Experiments suggest that the same choice of weights works beyond the case of quadratic f ; in fact the method seems to beat the competition even for nonconvex likelihoods*__
>
> We would like to clarify that the experiments are all about sampling from strongly logconcave distributions, as indicated in the paper.
>
> #########################################################################
>
> We have replied to each of the reviewer's comments, and we will appreciate it if the reviewer can seriously consider upgrading the score. Thank you.

---

> ### Author Response · Authors · 2022-11-28
> **Dear Reviewer GRbp**
>
> Dear Reviewer GRbp,
>
> We would very much appreciate it if the reviewer can engage with us, and kindly let us know if we have clarified all the concerns and provided satisfying answers. Or, let us know if we should address any parts in more detail, and we will be happy to do so. Thank you.
>
> All the best,
>
> The authors

---

> ### Author Response · Authors · 2022-12-05
> **Thanks for the feedback and for upgrading the score.**
>
> We saw that the reviewer has updated the review and given us some feedback. Thanks for the feedback and for upgrading the score.
>
>
> We agree that studying HMC for sampling from non-log-concave distributions is an important research direction, because there is little progress in sampling literature on establishing a non-asymptotic convergence rate of HMC when the potential is non-convex. However, the topic is disjoint from our work, as the reviewer kindly acknowledges.
>
> This work is concerned with sampling from strongly log-concave distributions via HMC,
> and therefore the 5 sampling tasks in our experiments are about sampling from strongly log-concave distributions. The most representative example of strongly log-concave distributions in the literature is Bayesian logistic regression to our knowledge. Therefore, in our experiments, one of the sampling tasks is Bayesian logistic regression. We have compared the performance of HMC with the proposed integration time and HMC with the best constant integration time for Bayesian logistic regression on three binary classification datasets. Another sampling task in our experiments is sampling from a mixture of two Gaussians, which is a strongly log-concave distribution when we confine the distance between the two Gaussians appropriately. We also consider sampling from a hard strongly log-concave distribution introduced by Lee et al. (2021).
> We believe that this work has included the most common tasks of sampling from strongly log-concave distributions in the literature.

---

### Author Response · Authors · 2022-11-15
**To all the reviewers**

We would highly appreciate it if the reviewers could read our response and let us know whether we have answered the questions and/or clarified the concerns. In the following, allow us to address and discuss our theoretical result of acceleration again.


The reviewers are correct that our theory currently only covers the quadratic case. However, we would like to point out that the quadratic case is an important setting that provides evidence whether we can apply the proposed technique with provable results. For example, in the paper we mentioned Chebyshev iteration and the work of Gradient descent with the Chebyshev step size (Agarwal et al. 2021), which also only covers the quadratic case.
Similarly, the classical paper of the Heavy Ball momentum method (Polyak 1964) ``only'' showed acceleration for minimizing strongly convex quadratic functions, but it became a foundation of further development both for theoretical results and practical algorithms.

Hamiltonian Monte Carlo (HMC) is a classical sampling method proposed more than three decades ago, which is very popular in practice. The question of how to accelerate sampling algorithms in parallel to optimization is an important open question that has received much recent attention, but not a concrete resolution yet.
We are very excited that our current work has made a solid step forward on accelerating this classical method. While our theoretical result is currently limited to the strongly convex quadratic potential, the experiments of sampling from general strongly log-concave distributions show that the proposed scheme of choosing the integration time accelerates HMC compared to the constant integration time. Hence, our work might help facilitate future research in accelerating HMC for sampling from general strongly log-concave distributions with strong theoretical guarantees, and we would like to pursue this direction in the future.


Specifically, we think that there are at least three directions to try
for going beyond quadratics.
- The first is to check if the analysis for the case of sampling from general strongly logconcave distributions in Chen and Vempala (2019) can be used/improved to get an accelerated rate.
- The second is to identify when and why gradient descent with the Chebyshev step sizes of Agarwal et al. (2021) has provable acceleration beyond quadratics for optimization,
because the intuition behind our acceleration result for HMC is based on a connection to this method. (The intuition can be found in our response to Reviewer XHWf.) Currently the acceleration result of Agarwal et al. (2021) is limited to minimizing strongly convex quadratic functions, even though they show promising empirical results compared to gradient descent with a constant step size for minimizing some non-quadratic smooth strongly convex functions. We think that identifying any conditions that gradient descent with the Chebyshev step sizes can have provable acceleration for minimizing general (or any subclasses of) smooth strongly convex functions might shed light on acceleration of HMC beyond quadratics.
- The third is to try analyzing ideal HMC for sampling from general strongly log-concave distributions in 1-dimension. While the differential equation of ideal HMC often does not have a closed form, which we believe is why showing acceleration beyond quadratics is challenging, it is interesting to check if we can get a bound that involves the cosine product as the case of quadratic potentials (perhaps modulo some factors). If so, then we could proceed to check if acceleration is possible.


Overall, we believe that our work may open doors to future theoretical works.


References:

[1] Naman Agarwal, Surbhi Goel, and Cyril Zhang. Acceleration via fractal learning rate schedules.
ICML, 2021.

[2] Boris Polyak. Some methods of speeding up the convergence of iteration methods. USSR Computational Mathematics and Mathematical Physics, 1964.

[3] Zongchen Chen and Santosh Vempala. Optimal convergence rate of Hamiltonian Monte Carlo for strongly logconcave distributions. International Conference on Randomization and Computation
(RANDOM), 2019.

---

### Decision · Program_Chairs · 2023-01-20

**Decision:**

Accept: poster

**Justification For Why Not Higher Score:**

The theory is limited to quadratic potential function (Gaussian case).

**Justification For Why Not Lower Score:**

It receives strong support from one reviewer

**Metareview: Summary, Strengths And Weaknesses:**

This paper proposes a scheme of time-varying integration based on Chebyshev polynomials for Hamiltoian Monte Carlo (HMC) methods. For the case of quadratic potential function, it achieves faster convergence in terms of Wasserstein-2 distance. The improvement on the dependence on condition number is akin to acceleration in optimization.

Strengths:

+This is the first accelerated result for HMC

+The use of Chebyshev polynomials in HMC is somewhat novel

Weaknesses:

-The theory is limited to quadratic potential function (Gaussian case).

After the author's response and reviewer discussion, the most negative reviewer has raised the score. Plus, this paper receives strong support from the other reviewer.  Thus, I recommend acceptance.


**Note From Pc:**

if the above contains the word "oral" or "spotlight" please see: "oral" presentation means -> notable-top-5% and "spotlight" means -> notable-top-25%. As stated in our emails, we are disassociating presentation type from AC recommendations

**Summary Of Ac-Reviewer Meeting:**

Before the AC-reviewer meeting, one of the reviewers who used to give 5 raised the score to 6.